# Perception-R1: Advancing Multimodal Reasoning Capabilities of MLLMs via Visual Perception Reward

**Tong Xiao**[1], **Xin Xu**[4], **Zhenya Huang**[1,2], **Hongyu Gao**[3],
**Quan Liu**[2,3], **Qi Liu**[1,2], **Enhong Chen**[1,2]*

[1]University of Science and Technology of China
[2]State Key Laboratory of Cognitive Intelligence
[3]iFLYTEK AI Research
[4]Hong Kong University of Science and Technology

## Abstract

Enhancing the multimodal reasoning capabilities of Multimodal Large Language Models (MLLMs) is a challenging task that has attracted increasing attention in the community. Recently, several studies have applied Reinforcement Learning with Verifiable Rewards (RLVR) to the multimodal domain in order to enhance the reasoning abilities of MLLMs. However, these works largely overlook the enhancement of multimodal perception capabilities in MLLMs, which serve as a core prerequisite and foundational component of complex multimodal reasoning. Through McNemar's test, we find that existing RLVR method fails to effectively enhance the multimodal perception capabilities of MLLMs, thereby limiting their further improvement in multimodal reasoning. To address this limitation, we propose Perception-R1, which introduces a novel visual perception reward that explicitly encourages MLLMs to perceive the visual content accurately, thereby can effectively incentivizing both their multimodal perception and reasoning capabilities. Specifically, we first collect textual visual annotations from the CoT trajectories of multimodal problems, which will serve as visual references for reward assignment. During RLVR training, we employ a judging LLM to assess the consistency between the visual annotations and the responses generated by MLLM, and assign the visual perception reward based on these consistency judgments. Extensive experiments on several multimodal math and general benchmarks demonstrate the effectiveness and robustness of our Perception-R1, which achieves superior performance on all benchmarks using only 1,442 training data. Our code and dataset will be available at https://github.com/tongxiao2002/Perception-R1.

## 1 Introduction

Multimodal reasoning is a fundamental capability for AI systems to solve complex real-world tasks and represents a critical step toward artificial general intelligence (AGI). Since the emergence of Multimodal Large Language Models (MLLMs), extensive studies have been proposed to enhance their multimodal reasoning capabilities. Early efforts focused on prompt engineering techniques (Zhang et al., 2023b; Zheng et al., 2023; Zhang et al., 2024a; Mitra et al., 2024; Xue et al., 2024), while subsequent works leveraged supervised fine-tuning (SFT) with high-quality reasoning datasets (Gao et al., 2024; Shi et al., 2024; Peng et al., 2024; Zhang et al., 2024c; Liu et al., 2024c). More recently, the success of reasoning models such as OpenAI-o1 (Jaech et al., 2024), DeepSeek-R1 (Guo et al., 2025), and Kimi-k1.5 (Team et al., 2025) has demonstrated the efficacy of large-scale reinforcement learning (RL), particularly RL with Verifiable Rewards (RLVR), in enhancing advanced reasoning capabilities of LLMs and MLLMs, thereby drawing significant attention from the research community.

Motivated by these pioneering works, a growing number of studies (Meng et al., 2025; Peng et al., 2025; Yang et al., 2025; Huang et al., 2025; Deng et al., 2025) have emerged to further advance the

---

*Corresponding author.

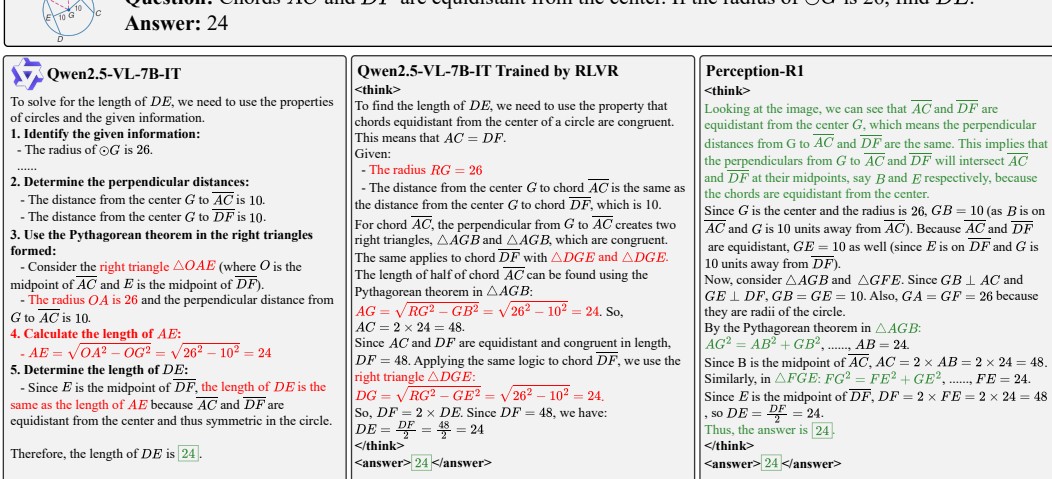

**Question:** Chords $\overline{AC}$ and $\overline{DF}$ are equidistant from the center. If the radius of $\odot G$ is 26, find $DE$.
**Answer:** 24

**Qwen2.5-VL-7B-IT**

To solve for the length of $DE$, we need to use the properties of circles and the given information.
**1. Identify the given information:**
 - The radius of $\odot G$ is 26.
 ......
**2. Determine the perpendicular distances:**
 - The distance from the center $G$ to $\overline{AC}$ is 10.
 - The distance from the center $G$ to $\overline{DF}$ is 10.
**3. Use the Pythagorean theorem in the right triangles formed:**
 - Consider the right triangle $\triangle OAE$ (where $O$ is the midpoint of $\overline{AC}$ and $E$ is the midpoint of $\overline{DF}$).
 - The radius $OA$ is 26 and the perpendicular distance from $G$ to $\overline{AC}$ is 10.
**4. Calculate the length of $AE$:**
 - $AE = \sqrt{OA^2 - OG^2} = \sqrt{26^2 - 10^2} = 24$
**5. Determine the length of $DE$:**
 - Since $E$ is the midpoint of $\overline{DF}$, the length of $DE$ is the same as the length of $AE$ because $\overline{AC}$ and $\overline{DF}$ are equidistant from the center and thus symmetric in the circle.

Therefore, the length of $DE$ is 24 .

**Qwen2.5-VL-7B-IT Trained by RLVR**
<think>
To find the length of $DE$, we need to use the property that chords equidistant from the center of a circle are congruent. This means that $AC = DF$.
Given:
 - The radius $RG = 26$
 - The distance from the center $G$ to chord $\overline{AC}$ is the same as the distance from the center $G$ to chord $\overline{DF}$, which is 10.
For chord $AC$, the perpendicular from $G$ to $\overline{AC}$ creates two right triangles, $\triangle AGB$ and $\triangle AGB$, which are congruent. The same applies to chord $\overline{DF}$ with $\triangle DGE$ and $\triangle DGE$.
The length of half of chord $\overline{AC}$ can be found using the Pythagorean theorem in $\triangle AGB$:
$AG = \sqrt{RG^2 - GB^2} = \sqrt{26^2 - 10^2} = 24$. So,
$AC = 2 \times 24 = 48$.
Since $AC$ and $DF$ are equidistant and congruent in length, $DF = 48$. Applying the same logic to chord $\overline{DF}$, we use the right triangle $\triangle DGE$:
$DG = \sqrt{RG^2 - GE^2} = \sqrt{26^2 - 10^2} = 24$.
So, $DF = 2 \times DE$. Since $DF = 48$, we have:
$DE = \frac{DF}{2} = \frac{48}{2} = 24$
</think>
<answer> 24 </answer>

**Perception-R1**
<think>
Looking at the image, we can see that $\overline{AC}$ and $\overline{DF}$ are equidistant from the center $G$, which means the perpendicular distances from G to $\overline{AC}$ and $\overline{DF}$ are the same. This implies that the perpendiculars from $G$ to $\overline{AC}$ and $\overline{DF}$ will intersect $\overline{AC}$ and $\overline{DF}$ at their midpoints, say $B$ and $E$ respectively, because the chords are equidistant from the center.
Since $G$ is the center and the radius is 26, $GB = 10$ (as $B$ is on $\overline{AC}$ and $G$ is 10 units away from $\overline{AC}$). Because $\overline{AC}$ and $\overline{DF}$ are equidistant, $GE = 10$ as well (since $E$ is on $\overline{DF}$ and $G$ is 10 units away from $\overline{DF}$).
Now, consider $\triangle AGB$ and $\triangle GFE$. Since $GB \perp AC$ and $GE \perp DF$, $GB = GE = 10$. Also, $GA = GF = 26$ because they are radii of the circle.
By the Pythagorean theorem in $\triangle AGB$:
$AG^2 = AB^2 + GB^2$, ......, $AB = 24$.
Since B is the midpoint of $\overline{AC}$, $AC = 2 \times AB = 2 \times 24 = 48$.
Similarly, in $\triangle FGE$: $FG^2 = FE^2 + GE^2$, ......, $FE = 24$.
Since $E$ is the midpoint of $\overline{DF}$, $DF = 2 \times FE = 2 \times 24 = 48$, so $DE = \frac{DF}{2} = 24$.
Thus, the answer is 24 .
</think>
<answer> 24 </answer>

Figure 1: A comparison of three MLLMs on a geometry problem. Both Qwen2.5-VL-7B-IT and its RLVR-trained variant make severe perception errors but manage to guess the answer, whereas our Perception-R1 first accurately describes the image and then solves the problem correctly.

multimodal reasoning capabilities of MLLMs leveraging RLVR, which train MLLMs solely with rewards derived from the correctness of the final answer. For instance, MM-Eureka (Meng et al., 2025) and R1-VL (Zhang et al., 2025) successfully stabilized the training of RLVR in the multimodal domain by incorporating additional training tricks and mechanisms. R1-OneVision (Yang et al., 2025) and Vision-R1 (Huang et al., 2025) first constructed a large-scale high-quality multimodal dataset by addressing the modality gap issue, then applied a training pipeline consisting of cold-start followed by RLVR to train MLLMs on the dataset. All these works effectively enhanced the multimodal reasoning capabilities of MLLMs, leading to substantial performance improvements.

Multimodal reasoning can be naturally decomposed into multimodal perception and logical reasoning (Amizadeh et al., 2020; Zhou et al., 2024), where multimodal perception is responsible for accurately understanding the multimodal input and supplying essential information for subsequent reasoning, thereby serving as the foundation for effective multimodal reasoning. Although RLVR-trained MLLMs demonstrate improved reasoning capabilities, our detailed analysis reveal that existing RLVR fails to effectively improve the multimodal perception capabilities of MLLMs, making it a major bottleneck that restricts their further advancement in multimodal reasoning.

For example, as illustrated in Figure 1, the original MLLM (left in Figure 1) makes severe multimodal perception errors (e.g., referring to "right triangle $\triangle OAE$" that does not exist in the image), indicating its limited multimodal perception capabilities. Nevertheless, it still manage to guess the correct answer. This makes existing RLVR method, which optimizes MLLMs solely based on answer accuracy, struggles to correct perception errors and may even reinforces this flawed reasoning path. Consequently, the resulting MLLM (middle in Figure 1) still exhibits weak multimodal perception capabilities similar to its original counterpart (e.g., referring to "$RG$" that does not exist), hindering the development of genuine multimodal reasoning capabilities.

We attribute this challenge to the rewards sparsity for multimodal perception when training MLLMs with existing RLVR, making it difficult to effectively enhance the multimodal perception capabilities of MLLMs. To address this challenge, we propose Perception-R1, which incorporates a novel and effective visual perception reward into the multimodal RLVR training process. The visual perception reward provides an additional reward signal beyond answer accuracy, explicitly encouraging MLLMs to perceive visual content accurately, thereby alleviating reward sparsity in RLVR training and facilitating more effective multimodal reasoning by strengthening the MLLMs' perceptual foundation.

Specifically, we introduce visual annotations into RLVR as auxiliary references, encouraging MLLMs to generate perception-accurate responses that closely align with them during training. To obtain such visual annotations, we first collect CoT trajectories with correct final answers from a state-of-the-art

multimodal reasoning model and then employ an LLM to extract natural language visual annotations from them. Our manual examination indicates that these visual annotations reach an accuracy of 96% (see Appendix B.2). During RLVR training, visual perception reward is assigned based on the consistency between the visual annotations and the responses of MLLM, as evaluated by a judging LLM via prompting. By incorporating the visual perception reward into RLVR, our Perception-R1 achieves the best performance compared to several strong baselines across most multimodal benchmarks using only 1,442 training samples, surpassing Vision-R1 (Huang et al., 2025), which requires 200K data samples for training in total.

In summary, our contributions are threefold:
(1). We investigate the behaviors of RLVR-trained MLLMs and their original counterparts, and find that their multimodal perception capabilities are not statistically significantly different, remaining a major bottleneck that limits further advancement in multimodal reasoning.
(2). We propose Perception-R1, which introduces a novel visual perception reward into RLVR. By providing an additional perception reward signal, Perception-R1 alleviates the reward sparsity in multimodal perception and effectively enhances the multimodal reasoning capabilities of MLLMs.
(3). Extensive experiments on several multimodal math and general benchmarks demonstrate the superiority of our Perception-R1, which exhibits significantly improved multimodal perception capabilities and achieves superior performance on all benchmarks using only 1,442 training samples.

## 2 RELATED WORK

### 2.1 MULTIMODAL LARGE LANGUAGE MODELS

Multimodal Large Language Models (MLLMs) have witnessed rapid advancements in recent years. Most studies (Bai et al., 2023; Wang et al., 2024b; Bai et al., 2025; Liu et al., 2023; Chen et al., 2024d) developed MLLMs by aligning a visual encoder to a pre-trained LLM through vision-language adaptors (VL-adaptors), making the modality alignment the core of MLLMs development. Early efforts focused on architectural designs to enhance alignment, exploring various forms of VL-adaptors and visual encoders. From the perspective of VL-adaptor, three mainstream types have been widely studied: cross-attention modules (Bai et al., 2023; Dai et al., 2023; Zhang et al., 2023a), parallel visual experts (Wang et al., 2024c; Dong et al., 2024) inspired by LoRA (Hu et al., 2022), and simple linear projection layers (Liu et al., 2023; Li et al., 2024; Wang et al., 2024b; Bai et al., 2025; Chen et al., 2024b). Among these, linear projection layers have demonstrated strong effectiveness (Laurençon et al., 2024) and are now predominantly adopted in SOTA MLLMs. Meanwhile, CLIP-based visual encoders have been found to possess intrinsic limitations in multimodal perception (Tong et al., 2024; Jiang et al., 2025), prompting the research of applying hybrid vision towers (Tong et al., 2024; Lu et al., 2024a) and scaling up the vision backbones (Chen et al., 2024d;c;b). Beyond the exploration of architecture of MLLMs, recent studies have also advanced modality alignment from a data perspective. Works such as LLaVA (Liu et al., 2023; 2024a;b; Li et al., 2024), Qwen-VL (Bai et al., 2023; Wang et al., 2024b; Bai et al., 2025), and InternVL (Chen et al., 2024d;c;b) have significantly scaled up both the volume and diversity of training data. For instance, LLaVA's training data grew from 753K samples to 9.36M in LLaVA-OV (Li et al., 2024), while the data diversity broadened from general images to include math reasoning, document and video understanding, substantially improving VL alignment and overall performance on multimodal benchmarks.

### 2.2 MULTIMODAL LARGE LANGUAGE MODELS REASONING

Since the advent of MLLMs, enhancing their complex multimodal reasoning capabilities has drawn increasing research attention. Early efforts (Gao et al., 2024; Shi et al., 2024; Peng et al., 2024; Xiao et al., 2024) focused on distilling CoT trajectories from proprietary models like GPT-4V (Achiam et al., 2023) and Gemini (Team et al., 2023) to inject reasoning abilities into open-source MLLMs. Although these methods can achieve improvement on targeted benchmarks, they lack generalizability to OOD domains. Motivated by OpenAI-o1's (Jaech et al., 2024) test-time scaling, many works (Xu et al., 2024; Xiang et al., 2024; Yao et al., 2024; Luo et al., 2025) explored the implementation of it in multimodal reasoning domain. Approaches like LLaVA-CoT (Xu et al., 2024), AtomThink (Xiang et al., 2024), and URSA (Luo et al., 2025) implemented o1-style reasoning by enforcing step-wise outputs and leveraging process-level rewards to evaluate intermediate steps. Recently, the remarkable success of DeepSeek-R1 (Guo et al., 2025) in improving LLM reasoning through large-scale RLVR

has motivated researchers to transfer similar approaches into the multimodal domain. Vision-RFT (Liu et al., 2025) and R1-V (Chen et al., 2025b) applied RLVR to object detection and counting tasks, significantly improving the image understanding capabilities of MLLMs. MM-Eureka (Meng et al., 2025), VLAA-Thinker (Chen et al., 2025a) and MMR1 (Leng* et al., 2025) extended RLVR on math reasoning tasks without cold-start, achieving substantial improvements in multimodal reasoning. In addition to standard RLVR, R1-VL (Zhang et al., 2025) and SophiaVL (Fan et al., 2025) incorporated additional rewards to further supervise the thinking process. Vision-R1 employed a pipeline that begins with a long CoT cold-start phase and subsequently conducts large-scale RL, leading to superior performance on several multimodal math benchmarks. Although prior works have made remarkable progress in enhancing the multimodal reasoning abilities of MLLMs, they overlooked the multimodal perception capabilities of MLLMs, which are essential for complex multimodal reasoning and remains difficult to optimize under existing RLVR method due to sparse rewards.

## 3 PRELIMINARIES

This section formulates the multimodal reasoning task (Section 3.1) and introduces key concepts of the RLVR algorithm (Section 3.2) employed in this work.

### 3.1 PROBLEM FORMULATION

In this work, we investigate the multimodal reasoning task in the context of MLLMs. Let $\mathcal{D} = (x_1, x_2, ..., x_N)$ be a multimodal reasoning dataset, where each data sample $x_i = (V, Q, a)$ comprises visual input $V$ (e.g., image), a textual query $Q$, and the corresponding ground-truth answer $a$. The multimodal reasoning task is defined as follows: given a data sample $x_i \in \mathcal{D}$ as input, the MLLM is required to generate a textual token sequence $y$ that aims to reach the ground-truth answer $a$.

### 3.2 REINFORCEMENT LEARNING WITH VERIFIABLE REWARDS (RLVR)

Reinforcement Learning with Verifiable Rewards (RLVR) is an RL variation that eliminates the dependency on external reward models by using ground-truth answers for reward assignment, which both mitigates challenging reward hacking issues (Denison et al., 2024) and substantially reduces computational overhead. Existing methods (Guo et al., 2025; Meng et al., 2025; Huang et al., 2025) typically apply RLVR with two main components, including reward functions and GRPO algorithm.

**Reward Functions:** The reward functions consist of two components:
(1). *Format Reward* ($r_f$) encourages MLLMs to generate in a structured "think-then-answer" format, with the reasoning process enclosed in `<think>` tags and the answer enclosed in `<answer>` tags.
(2). *Accuracy Reward* ($r_a$) drives the reasoning optimization in RLVR training by evaluating the correctness of predicted answer. Existing works (Face, 2025; Huang et al., 2025) mostly adopt a symbolic system to judge the equivalence of ground-truth $a$ and answer in MLLMs' response $y$.

Since format reward $r_f$ only enforces structured output, while accuracy reward $r_a$ plays a central role in enhancing the multimodal reasoning capabilities of MLLMs, we refer to RLVR with a following reward function as **accuracy-only RLVR**:

$$r(y_i, a) = \alpha \cdot r_f(y_i) + \beta \cdot r_a(y_i, a) \tag{1}$$

where $\alpha, \beta$ are coefficients that control the impact of these two rewards.

**Group Relative Policy Optimization (GRPO)** (Shao et al., 2024) is a variant of Proximal Policy Optimization (PPO) (Schulman et al., 2017), which eliminates the need for a critic by estimating baseline rewards from groups of rollouts, thereby reducing computational overhead while maintaining performance. For each data sample $x \in \mathcal{D}$, GRPO first samples a group of rollouts $Y = (y_1, y_2, ..., y_G)$ from the policy model $\pi_\theta$, then computes advantage $\hat{A}_i$ by normalizing rewards across these rollouts:

$$\hat{A}_i = \frac{r(y_i, a) - \text{mean}\{r(y_1, a), r(y_2, a), ..., r(y_G, a)\}}{\text{std}\{r(y_1, a), r(y_2, a), ..., r(y_G, a)\}} \tag{2}$$

After obtaining the advantages, GRPO optimizes the policy model $\pi_\theta$ by maximizing the objective:

$$\mathcal{J}(\theta) = \mathbb{E}_{x \in \mathcal{D}, \{y_i\}_{i=1}^G \sim \pi_{\theta_{\text{old}}}}$$

$$\left[ \frac{1}{G} \sum_{i=1}^G \frac{1}{|y_i|} \sum_{t=1}^{|y_i|} \left\{ \min \left[ s_t(x, y_i)\hat{A}_i, \text{clip}\left(s_t(x, y_i), 1 - \varepsilon, 1 + \varepsilon\right)\hat{A}_i \right] - \delta \cdot \text{KL}(\pi_\theta \| \pi_{ref}) \right\} \right] \quad (3)$$

where $s_t(x, y_i) = \frac{\pi_\theta(y_{i,t}|x, y_{i,<t})}{\pi_{\theta_{\text{old}}}(y_{i,t}|x, y_{i,<t})}$, $\varepsilon$ is the clipping hyper-parameter, $\delta$ is the coefficient that controls the impact of Kullback-Leibler (KL) divergence, $\pi_\theta$ is the policy model and $\pi_{ref}$ is the fixed reference model that is usually initialized from the initial policy.

## 4 METHODS

### 4.1 ACCURACY-ONLY RLVR FAILS TO IMPROVE MULTIMODAL PERCEPTION IN MLLMS

Multimodal reasoning capabilities can be naturally decomposed into multimodal perception and logical reasoning capabilities (Amizadeh et al., 2020; Zhou et al., 2024). Although RLVR has been proven effective in enhancing logical reasoning, many failure cases similar to Figure 1 reveal its limited impact on enhancing multimodal perception. To further validate this observation, we first train Qwen2-VL-7B-IT (Wang et al., 2024b) and Qwen2.5-VL-7B-IT (Bai et al., 2025) on Geometry3K (Lu et al., 2021a) dataset using accuracy-only RLVR, and then conduct investigations by analyzing CoT trajectories on MathVista (Lu et al., 2024b) and MathVerse (Zhang et al., 2024b).

Our investigation yields the following results (Further details are provided in Appendix B.1):
(1). For Qwen2-VL-7B-IT, we analyze 50 and 25 incorrect cases from MathVista and MathVerse, respectively, and find that 72% and 68% of these failures are caused by multimodal perception errors. For Qwen2.5-VL-7B-IT, the corresponding proportions are 78% and 76%, respectively. These results highlight that multimodal perception remains a major bottleneck for RLVR-trained MLLMs, which limits their further advancement in multimodal reasoning.
(2). We conduct exact binomial variation of McNemar's test (McNemar, 1947; Edwards, 1948) on 50 multimodal problems randomly sampled from MathVista. For Qwen2-VL-7B-IT, the numbers of discordant cases related to multimodal perception are 1 and 5, respectively. For Qwen2.5-VL-7B-IT, the numbers are 2 and 4, respectively. As a result, the exact binomial test yields $p$-values of $0.22$ and $0.69$, both far above the $0.05$ significance level, indicating that the multimodal perception abilities of the accuracy-only RLVR trained MLLMs do not significantly differ from those of the base model.

### 4.2 PERCEPTION-R1

We attribute this limitation to the reward sparsity of accuracy-only RLVR, as answer correctness does not guarantee accurate multimodal perception, (e.g., as illustrated in Figure 1), making it difficult for accuracy-only RLVR to effectively optimize the multimodal perception capabilities of MLLMs. To tackle this issue, we propose Perception-R1, which introduces a novel and effective visual perception reward into RLVR, explicitly guiding MLLMs toward improving their multimodal perception capabilities, thereby effectively enhancing their overall multimodal reasoning performance.

Since directly introducing a multimodal reward model may introduce additional reward hacking issues, we largely adhere to the RLVR paradigm in designing the visual perception reward. As shown in Figure 2, we first collect CoT trajectories that contain accurate visual information and then extract visual annotations from them. These visual annotations serve as references for assigning visual perception reward, analogous to the use of ground-truth answers in computing accuracy reward. Subsequently, a judging LLM is used to assess the consistency between visual annotations and the MLLM generated responses, thereby assisting in the assignment of the visual perception reward. Finally, we aggregate all rewards and apply GRPO to optimize the policy model.

#### 4.2.1 CURATION OF VISUAL ANNOTATIONS

Visual images often encode rich and complex information that is difficult to convey fully through text. Since our ultimate objective is to enhance the multimodal reasoning capabilities of MLLMs rather than to generate faithful image captions, we focus on guiding MLLMs to concentrate on visual

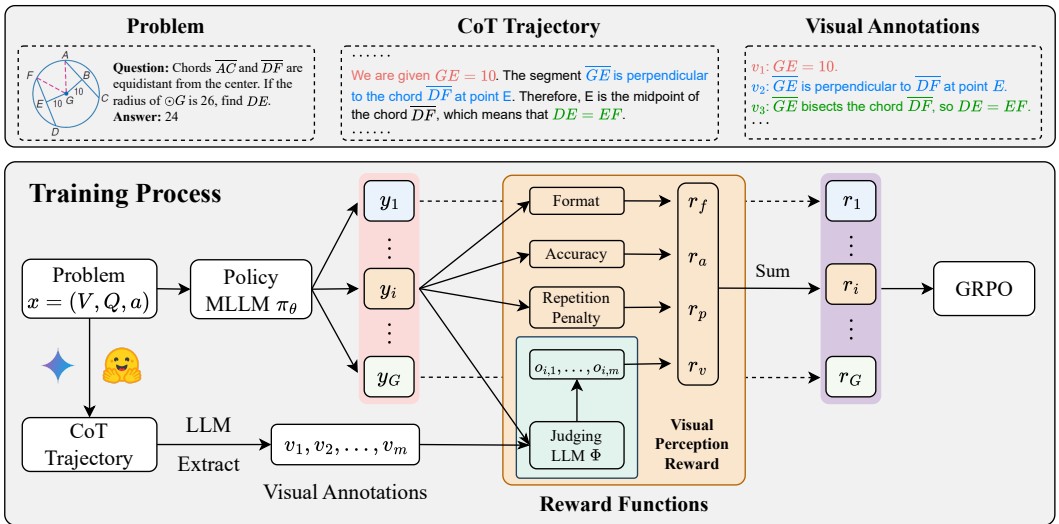

Figure 2: Overview of training pipeline of the proposed Perception-R1. In addition to the accuracy and format rewards, we introduce a novel visual perception reward that explicitly guides MLLMs toward improving their multimodal perception capabilities.

content pertinent to problem solving, such as identifying $GE = 10$ rather than being influenced by superficial cues like line color in Figure 2.

To obtain such visual information, we employ a SOTA proprietary MLLM to generate CoT trajectories on multimodal reasoning dataset $\mathcal{D}$, treating the visual information embedded within these trajectories as accurate and highly relevant to problem-solving. Notably, these CoT trajectories can also be obtained from existing open-source multimodal SFT datasets. We then further prompt a strong text-only LLM to extract this embedded visual information from each CoT trajectory into a sequence of visual annotations $\mathcal{V} = (v_1, v_2, ..., v_m)$, where each $v_i$ represents a textual atomic visual annotation of the image that is critical for problem-solving (e.g., $GE = 10$, $\overline{GE} \perp \overline{DF}$ in Figure 2), and $m$ denotes the total number of visual annotations within the trajectory. These visual annotations $\mathcal{V}$ will serve as ground-truth references for evaluating whether the policy model accurately perceives visual content during RLVR training, analogous to the role of ground-truth answers in the accuracy reward.

### 4.2.2 VISUAL PERCEPTION REWARD

During RLVR training, we need to evaluate the consistency between the visual annotations $\mathcal{V}$ and the visual description embedded in the responses generated by the policy model $\pi_\theta$. Since symbolic systems struggle to capture the complex semantics of natural language, we address this limitation by introducing a judging LLM $\Phi$ to assess whether each atomic visual annotation $v_i$ is accurately reflected in the responses generated by policy model, thereby extending the source of reward signals.

Formally, given a data sample $x \in \mathcal{D}$ and its corresponding visual annotations $\mathcal{V} = (v_1, v_2, ..., v_m)$, we first sample a response $y_i$ from the policy model $\pi_\theta$, then employ a judging LLM $\Phi$ to assess whether each atom annotation $v_j$ is presented in $y_i$. Consequently, this process results in a judgment sequence $\mathcal{J} = (o_{i,1}, o_{i,2}, ..., o_{i,m})$, where $o_{i,j} \in \{0, 1\}$ indicates whether $v_j$ is accurately reflected in $y_i$ or not. Obtaining $\mathcal{J}$, we can compute the visual perception reward $r_v$ for $y_i$:

$$r_v(y_i, \mathcal{V}) = \frac{\text{sum}\{o_{i,1}, o_{i,2}, ..., o_{i,m}\}}{|o_{i,1}, o_{i,2}, ..., o_{i,m}|}, \text{ where } o_{i,j} = \Phi(y_i, v_j) \in \{0, 1\}, v_j \in \mathcal{V} \quad (4)$$

Accordingly, our visual-enhanced reward function is defined as follows:

$$r(y_i, a, \mathcal{V}) = \alpha \cdot r_f(y_i) + \beta \cdot r_a(y_i, a) + \gamma \cdot r_v(y_i, \mathcal{V}) + r_p(y_i) \quad (5)$$

where $r_f$ and $r_a$ are format reward and accuracy reward explained in Section 3.2, $\gamma$ is the coefficient that controls the impact of visual perception reward, $r_p$ is the repetition penalty reward that discourage repetitive behavior during MLLMs' generation. The introduction of $r_p$ is motivated by our observation

Table 1: Performance comparison between Perception-R1 and baselines on 8 benchmarks. The best and second-best results of Open-Source Reasoning MLLMs are highlighted in red and blue. † R1-VL-7B and Vision-R1-7B both trained on WeMath and MathVision, their results are omitted.

| Model | #Data | Math Benchmarks | | | | General Benchmarks | | | |
|---|---|---|---|---|---|---|---|---|---|
| | | MathVista testmini | MathVerse testmini | MathVision test | WeMath testmini | MMMU val | MMMU-Pro overall | MMStar val | EMMA full |
| *Proprietary MLLMs* | | | | | | | | | |
| GPT-4o | / | 63.8 | 50.2 | 30.4 | 68.8 | 69.1 | 51.9 | - | 32.7 |
| Claude-3.7-Sonnet | / | 66.8 | 52.0 | 41.3 | 72.6 | 71.0 | 51.5 | 65.1 | 35.1 |
| OpenAI-o1 | / | 73.9 | 57.0 | 60.3 | - | 78.2 | 62.4 | - | 45.7 |
| *Open Source General MLLMs* | | | | | | | | | |
| Qwen2-VL-7B-IT | / | 58.6 | 31.1 | 16.7 | 42.3 | 46.9 | 29.6 | 56.0 | 24.5 |
| Qwen2.5-VL-7B-IT | / | 68.1 | 47.4 | 25.1 | 61.4 | 55.2 | 37.0 | 63.1 | 24.9 |
| Qwen2.5-VL-72B-IT | / | 75.8 | 55.8 | 37.9 | 71.9 | 70.2 | 49.5 | 70.8 | 38.2 |
| InternVL2.5-8B | / | 64.4 | 39.5 | 19.7 | 53.5 | 56.0 | 34.3 | 62.8 | - |
| *Open-Source Reasoning MLLMs* | | | | | | | | | |
| URSA-7B | 3.06M | 59.8 | 45.7 | - | - | - | - | - | - |
| R1-VL-7B | 260K | 62.7 | 40.8 | -† | -† | 52.3 | 29.4 | 56.7 | 23.5 |
| R1-OneVision-7B | 155K | 65.0 | 46.5 | 21.9 | 61.9 | 52.9 | 33.8 | 58.9 | 23.6 |
| OpenVLThinker-7B | 25K | 71.3 | 47.4 | 24.3 | 66.3 | 58.4 | 37.8 | 63.8 | 27.0 |
| VLAA-Thinker-7B | 25K | 70.7 | 51.2 | 26.7 | 66.3 | 54.7 | 37.2 | 62.7 | 26.6 |
| SophiaVL-R1-7B | 130K | 70.6 | 49.0 | 26.6 | 64.8 | 56.7 | 38.8 | 63.1 | 27.4 |
| MM-Eureka-7B | 15K | 72.5 | 51.9 | 27.6 | 65.6 | 58.0 | 38.3 | 64.2 | 28.1 |
| Vision-R1-7B | 200K | 73.1 | 52.4 | -† | -† | 55.2 | 37.6 | 62.6 | **28.2** |
| **Perception-R1-7B** | **1.4K** | **74.2** | **54.3** | **28.6** | **72.0** | **60.8** | **42.4** | **64.5** | 27.5 |

that directly incorporating $r_v$ will result in increased repetition in the generated responses, which in turn impairs the model's multimodal reasoning capabilities. Following prior works (Yeo et al., 2025; Face, 2025), we implement $r_p$ using a simple $N$-gram repetition penalty.

During RLVR training, we replace the reward function $r(y_i, a)$ in Eq.1 by our visual-enhanced reward $r(y_i, a, \mathcal{V})$, and train the MLLM to maximize the GRPO objective exhibited in Eq.3.

# 5 EXPERIMENTS

## 5.1 EXPERIMENT SETTINGS

**Training Dataset.** We adopt Geometry3K (Lu et al., 2021a) dataset as our training data, which originally contains 2,101 samples for training. To obtain the visual annotations, we employ Gemini-2.5-Pro (Team et al., 2023) to generate CoT trajectories on the training data and retain those with correct answers. We then use Qwen2.5-32B-IT (Yang et al., 2024) to extract visual annotations from the retained CoT trajectories. This process results in a total of 1,442 data samples with associated visual annotations. Model training settings can be found in Appendix C.1.

**Benchmarks and Evaluation Settings.** For comprehensive evaluation, we evaluate Perception-R1 on a variety of challenging multimodal benchmarks, covering both math and general domains. The math benchmarks include MathVista (Lu et al., 2024b), MathVerse (Zhang et al., 2024b), MathVision (Wang et al., 2024a) and WeMath (Qiao et al., 2024). The general benchmarks comprise MMMU (Yue et al., 2024a), MMMU-Pro (Yue et al., 2024b), MMStar (Chen et al., 2024a) and EMMA (Hao et al., 2025). During inference, we use vLLM (Kwon et al., 2023) for efficiency and apply greedy decoding with a temperature of 0.0.

**Baselines.** We compare our method against several powerful MLLMs: **(1) Proprietary MLLMs:** GPT-4o (Hurst et al., 2024), OpenAI-o1 (Jaech et al., 2024), Claude-3.7-Sonnet (Anthropic, 2024), **(2) Open-Source General MLLMs:** Qwen2-VL-7B-IT (Wang et al., 2024b), Qwen2.5-VL-7B-IT, Qwen2.5-VL-72B-IT (Bai et al., 2025), InternVL2.5-8B (Chen et al., 2024b), **(3) Open-Source Reasoning MLLMs:** URSA-7B (Luo et al., 2025), R1-OneVision (Yang et al., 2025), R1-VL (Zhang et al., 2025), OpenVLThinker (Deng et al., 2025), VLAA-Thinker (Chen et al., 2025a), SophiaVL-R1-7B (Fan et al., 2025), MM-Eureka (Meng et al., 2025), Vision-R1 (Huang et al., 2025).

## 5.2 MAIN RESULTS

We present the performance comparison between our Perception-R1 and existing powerful methods across 8 mainstream multimodal benchmarks in Table 1. The performance of applying our method on Qwen2-VL-7B-IT is presented in Appendix B.3. We summarize our findings as follows:

**Perception-R1 achieves the best performance on most of the benchmarks.** As demonstrated in the table, despite being trained on a small dataset of only 1,442 samples, our Perception-R1 still achieves remarkable performance across all benchmarks, outperforming previous powerful methods on all benchmarks except EMMA. We also conduct statistical significance testing using a one-sample t-test, finding that the average improvement is significant with $p < 0.01$ compared to Vision-R1-7B and MM-Eureka-7B. This result provides strong evidence for the superior performance of our proposed Perception-R1. It also underscores the critical role of multimodal perception in enabling effective multimodal reasoning, suggesting that accuracy-only RLVR requires further adaptation when applied to the multimodal reasoning domain. Although Perception-R1 is trained on a mere 1,442 math geometry problems, it still achieves the best performance across several general benchmarks. It not only highlights Perception-R1's superior robustness and generalizability but also demonstrates the critical role of multimodal perception in multimodal reasoning and the rationality of our motivation.

**The multimodal perception capabilities of Perception-R1 show tangible improvements.** In addition to the overall performance of Perception-R1 on benchmarks presented in Table 1, we provide further evidence for the significant improvement of Perception-R1 in multimodal perception capabilities from the following two aspects: (1). We present the performance of Perception-R1 and representative baselines on the Vision-Only subsets of the MathVerse and MMMU-Pro benchmarks in Table 12. These subsets exclusively accept images as input, thereby posing a more rigorous challenge to the multimodal perception capabilities of MLLMs. As shown in the table, our Perception-R1 still achieves the best performance and outperforms baselines by a large margin, which strongly validates the superior multimodal perception capabilities of Perception-R1. (2). Similar to statistical test in Section 4.1, we also conduct McNemar's test on Perception-R1. We investigate the same 50 problems as presented in Section 4.1 and find that the numbers of discordant cases for multimodal perception are 2 and 10, respectively. As a result, the exact binomial variation of McNemar's test (McNemar, 1947) yields exact $p$ value of 0.04, below the 0.05 significance threshold, indicating that the multimodal perception capabilities of Perception-R1 is substantially improved compared to the original MLLM.

**Perception-R1 effectively enhances multimodal reasoning capabilities of MLLMs in a highly data-efficient manner.** Although existing methods such as MM-Eureka (Meng et al., 2025) and Vision-R1 (Huang et al., 2025) have demonstrated strong data efficiency in enhancing the multimodal reasoning capabilities of MLLMs compared to prior SOTA SFT and PRM approach (Luo et al., 2025), our Perception-R1 achieves even better performance using over $100\times$ less data than Vision-R1 and $10\times$ less data than MM-Eureka, demonstrating its exceptional data efficiency in developing reasoning MLLMs. This finding suggests that data efficiency can be substantially improved by incorporating richer reward signals from data beyond the final answer, as demonstrated by our proposed visual perception rewards. We believe Perception-R1 will achieve further enhanced performance when more high-quality and high-diversity training data is incorporated into its training process in the future.

## 5.3 ABLATION STUDY

In this section, we conduct ablation studies from two perspectives: (1) evaluating the effectiveness of each component of Perception-R1, i.e., the visual perception reward and the repetition penalty; and (2) comparing Perception-R1 with alternative approaches, including directly using an MLLM as the reward model and employing supervised fine-tuning to train the base model.

We present the results of ablation studies in Table 2. As shown in the table, firstly, the accuracy across all benchmarks declines when either the visual perception reward or the repetition penalty is removed, demonstrating the effectiveness and necessity of both components in our Perception-R1. Secondly, all ablations incorporating visual perception reward outperform others that are trained with accuracy-only RLVR on the "Vision Only" (VO) subset of MathVerse, further indicating that our proposed visual perception reward enhances the multimodal perception capabilities of MLLMs. Thirdly, directly employing a powerful MLLM (Qwen2.5-VL-32B-IT) as the reward model does not yield better performance than our Perception-R1. We attribute this to reward hacking (See Appendix B.6), which underscores the importance of constructing verifiable visual annotations. To

Table 2: Component & approach ablation studies of Perception-R1. The best result is marked in red.

| Model | Math Benchmarks | | | | General Benchmarks | | | |
|---|---|---|---|---|---|---|---|---|
| | MathVista testmini | MathVerse testmini | MathVision test | WeMath testmini | MMMU val | MMMU-Pro overall | MMStar val | EMMA full |
| Qwen2.5-VL-7B-IT | 68.1 | 47.4 | 25.1 | 61.4 | 55.2 | 37.0 | 60.2 | 24.9 |
| + GRPO | 73.3 | 51.3 | 26.6 | 69.5 | 58.0 | 38.2 | 63.1 | 24.9 |
| **Perception-R1-7B** | **74.2** | **54.3** | **28.6** | **72.0** | **60.8** | **42.4** | **64.5** | 27.5 |
| *Component Ablation* | | | | | | | | |
| w/o Visual Perception Reward | 73.6 | 53.0 | 27.6 | 70.4 | 57.2 | 40.1 | 63.5 | 27.9 |
| w/o Repetition Penalty | 73.6 | 52.6 | 26.9 | 68.5 | 59.1 | 40.6 | 63.6 | 27.6 |
| *Approach Ablation* | | | | | | | | |
| Qwen2.5-VL-7B-IT + SFT | 67.3 | 39.1 | 21.3 | 49.1 | 52.8 | 35.2 | 59.6 | **28.3** |
| Qwen2.5-VL-32B-IT as RM | 73.2 | 54.1 | 26.8 | 66.3 | 58.9 | 40.6 | 61.7 | 26.6 |

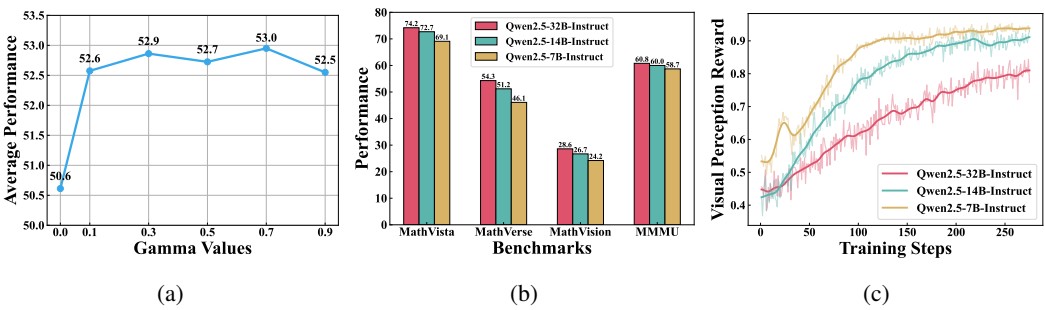

Figure 3: (a). Average performance across all benchmarks with varying $\gamma$ values. (b). Comparison of performance across benchmarks when using different judging LLMs. (c). Dynamics of visual perception reward during training when using different judging LLMs.

demonstrate the effectiveness of our overall RL training pipeline, we also conduct SFT experiment on the base model using the same 1,442 CoT trajectories distilled from Gemini-2.5-Pro (Team et al., 2023). From Table 2, it is observed that the SFT model yields inferior performance, with results on most benchmarks falling short of those of the base model after training. This phenomenon highlights the superior generalization ability and data efficiency of Perception-R1 compared to SFT method.

## 5.4 FURTHER ANALYSIS OF VISUAL PERCEPTION REWARD

To further explore the dynamics of the visual perception reward, we conduct experiments by varying the coefficient $\gamma$ in Eq. 5 and evaluating the impact of different judging LLMs.

To study the impact of coefficient $\gamma$, we train a series of models with $\gamma \in \{0, 0.1, 0.3, 0.5, 0.7, 0.9\}$, and present their average performance across all benchmarks in Figure 3a. We observe that models trained with different values of $\gamma$ achieve comparable performance across all benchmarks, while all significantly outperform the model that does not incorporate the visual perception reward. The result suggests that only a small amount of visual optimization signal is sufficient to effectively incentivize the multimodal perception and reasoning capabilities of MLLMs, and increasing the value of $\gamma$ does not lead to significantly better results. We attribute this to GRPO, which normalizes the advantages across responses that receive different visual perception rewards when other rewards are identical.

Given that the judging LLM plays a central role in assigning visual perception rewards, we investigate how its capability affects the performance of the resulting MLLM. Specifically, we employ Qwen2.5 (Yang et al., 2024) models of varying sizes as judging LLMs to train Qwen2.5-VL-7B-IT with $\gamma$ fixed at 0.7. The performance of the resulting models across benchmarks is presented in Figure 3b. As the capabilities of the judging LLMs decrease, the performance of the resulting MLLM consistently deteriorates, with the model trained using the 7B judging LLM even underperforming the original MLLM on MathVerse (46.1% vs. 47.4%) and MathVision (24.2% vs. 25.1%). According to the training dynamics of visual perception reward presented in Figure 3c, the reward increases rapidly and saturates early when using weak judging LLMs, implying the presence of severe reward hacking issues that misguide the resulting MLLM away from accurate problem solving.

## 6 CONCLUSION

In this paper, we first conduct McNemar's test on accuracy-only RLVR-trained MLLMs and find no statistically significant improvement in their multimodal perception capabilities compared to their original counterparts, which consequently limits their further advancement in multimodal reasoning. To address this limitation, we propose Perception-R1, which introduces a novel visual perception reward in addition to the standard accuracy reward, explicitly encouraging accurate visual perception during RLVR training. Specifically, we first collect textual visual annotations from CoT trajectories as references, and then assign visual perception reward by evaluating the consistency between these annotations and MLLM-generated response using a judging LLM. Extensive experiments demonstrate the effectiveness of Perception-R1, achieving the best performance compared to multiple baselines on most multimodal math and general benchmarks using only 1,442 training samples.

## REPRODUCIBILITY STATEMENT

To ensure the reproducibility of our work, we made efforts in three key areas: providing a clear methodological description, detailing core implementation configurations, and releasing the source code, datasets, and model checkpoints as open-source resources. In Section 4.2, we present a comprehensive description of the implementation of the visual perception reward and Perception-R1. The prompts used to obtain visual annotations $\mathcal{V}$ and to judge the consistency between the policy model's response and $\mathcal{V}$ are provided in Appendix C.2. Detailed training configurations are listed in Appendix C.1. To further support full reproducibility, we include our dataset and training code for Perception-R1 in the supplementary materials, and we will release the dataset, source code, and model checkpoint to the community upon publication.

## ACKNOWLEDGMENTS

This research was supported by the National Science and Technology Major Project (No.2023ZD0121102), the National Natural Science Foundation of China (No.U23A20319, No.62477044), Anhui Provincial Natural Science Foundation (No.2308085QF229), the support of the Young Elite Scientists Sponsorship Program by CAST (No.2024QNRC001), and the Fundamental Research Funds for the Central Universities (No.WK2150110038).

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

# A BENCHMARKS AND BASELINES

## A.1 BENCHMARKS

- **MathVista** (Lu et al., 2024b) MathVista is a consolidated benchmark for multimodal mathematical reasoning. We evaluate our Perception-R1 and all baselines on `testmini` split of MathVista, which consists of five subtasks: Textbook Question Answering, Visual Question Answering, Geometry Problem Solving, Math Word Problems and Figure Question Answering.

- **MathVerse** (Zhang et al., 2024b) MathVerse is a benchmark designed to evaluate the reasoning capabilities of MLLMs under varying proportions of textual and visual information. We evaluate our Perception-R1 and all baselines on `testmini` split of MathVerse, which includes 5 subset: Text Dominant (TD), Text Lite (TL), Vision Intensive (VI), Vision Dominant (VD), and Vision Only (VO). MathVerse covers three subtasks: "Plane Geometry" with 2,550 problems, "Functions" with 795 problems and "Solid Geometry" with 595 problems.

- **MathVision** (Wang et al., 2024a) MathVision consists of 3,040 high quality mathematical problems with visual contexts sourced from real math competitions. We evaluate our Perception-R1 and all baselines on `test` split of MathVision, which includes five difficulty levels and 16 subtasks.

- **WeMath** (Qiao et al., 2024) WeMath is the benchmark specifically designed to explore the problem-solving principles beyond the end-to-end performance, spanning 67 hierarchical knowledge concepts and 5 layers of knowledge granularity. We evaluate our Perception-R1 and all baselines on `testmini` split of WeMath under the multiple-choice setting.

- **MMMU** (Yue et al., 2024a) MMMU is a widely used multi-discipline multimodal benchmark that covers a broad scope of tasks, including Art, Business, Health & Medicine, Science, Humanities & Social Science, and Tech & Engineering, and over subfields, thus can comprehensively assess the multimodal reasoning abilities of a MLLM. We evaluate our Perception-R1 and all baselines on `val` subset of MMMU.

- **MMMU-Pro** (Yue et al., 2024b) MMMU-Pro is a more robust version of MMMU benchmark. MMMU-Pro improves MMMU from following 3 perspectives: (1). Excluding the problem that can be answered by text-only models, (2). augmenting the candidate options of multiple choice problems, making guessing more infeasible, and (3). introducing vision-only input setting where questions are embedded within images.

- **MMStar** (Chen et al., 2024a) MMStar comprises 1,500 meticulously curated problems sourced from a diverse range of existing multimodal benchmarks. To guarantee exceptional quality, the selection of problems for MMStar adheres to two core principles: (1). Visual information must be indispensable to solving the problem, and (2). avoiding data leakage.

- **EMMA** (Hao et al., 2025) EMMA is also a multi-discipline multimodal benchmark that covers mathematics, physics, chemistry, and coding. Different from previous multimodal benchmarks, EMMA emphasize the importance of organically reason over and with both text and images, therefore places higher requirements on the multimodal reasoning capabilities of MLLMs.

## A.2 BASELINES

- **URSA-7B** (Luo et al., 2025) URSA enhanced the multimodal reasoning capabilities of MLLMs through an SFT approach. It employed a three-part data synthesis strategy to construct a high-quality CoT reasoning dataset for SFT. USRA further incorporated a dual-view trajectory labeling approach, resulting in the DualMath-1.1M dataset, and trained a PRM to achieve test-time scaling.

- **R1-VL-7B** (Zhang et al., 2025) R1-VL proposed a new online reinforcement learning framework StepGRPO, which enabled MLLMs to self-improve reasoning ability via simple, effective and dense step-wise rewarding. It consisted of two dense reasoning rewards: StepRAR and StepRVR. StepRAR was used to reward the accurate intermediate reasoning steps and StepRVR was used to reward the well-structure of the overall reasoning path.

- **R1-OneVision-7B** (Yang et al., 2025) R1-OneVision adopted a cold-start then RL training pipeline to enhance the reasoning capabilities of MLLMs. It first addressed the modality gap to construct a

Table 3: Confusion matrix of Qwen2-VL-7B-IT evaluated on $\mathcal{D}_e$.

|                    | Correct Answer | Wrong Answer |
| ------------------ | -------------- | ------------ |
| Correct Perception | 15             | 4            |
| Wrong Perception   | 4              | 27           |

Table 4: Confusion matrix of accuracy-only RLVR trained Qwen2-VL-7B-IT evaluated on $\mathcal{D}_e$.

|                    | Correct Answer | Wrong Answer |
| ------------------ | -------------- | ------------ |
| Correct Perception | 23             | 0            |
| Wrong Perception   | 5              | 22           |

high-quality long CoT multimodal dataset for cold-start initialization, then applied accuracy-only RLVR on 10K randomly sampled data to further incentivize MLLM's reasoning abilities.

- **OpenVLThinker-7B** (Deng et al., 2025) OpenVLThinker adopted an approach that iteratively leverages SFT on lightweight training data and RL to improve reasoning capabilities of MLLMs. During the training pipeline, OpenVLThinker progressively evolved the data across iterations, retaining more challenging examples for later stages of training.

- **VLAA-Thinker-7B** (Chen et al., 2025a) VLAA-Thinker was developed by directly conducting RL on the VLAA-Thinking-RL-25K dataset using Qwen2.5-VL models. The main contributions of VLAA-Thinker were twofold: (1). VLAA-Thinking-RL-25K dataset was constructed by carefully selecting multimodal data from a variety of existing multimodal datasets. (2). It proposed a mixed reward approach to train MLLM during RL.

- **SophiaVL-R1-7B** (Fan et al., 2025) SophiaVL-R1 argued that outcome-based rewards alone cannot ensure a high-quality thinking process. To address this, it first trained a thinking reward model to evaluate the reasoning quality of intermediate steps. This reward model was then incorporated into RL to provide an additional thinking reward, guiding the policy model to generate trajectories with more coherent and well-reasoned intermediate steps.

- **MM-Eureka-7B** (Meng et al., 2025) MM-Eureka constructed the MMK12 dataset, which contained 15,616 high quality multimodal reasoning data, and then directly applied accuracy-only RLVR on this dataset. To stabilize and improve training, MM-Eureka incorporated several techniques during RL, including online data filtering, the removal of KL penalty, and a two-stage training strategy.

- **Vision-R1-7B** (Huang et al., 2025) Vision-R1 adopted a two-stage pipeline consisting of cold-start initialization followed by reinforcement learning to enhance the multimodal reasoning capabilities of MLLMs. It first employed powerfull MLLMs and DeepSeek-R1 (Guo et al., 2025) to fill the modality gap and curate 200K multimodal CoT data for cold-start initialization. In the second stage, Vision-R1 applied accuracy-only RLVR on an additional 10K math problems, incorporating the proposed Progressive Thinking Suppression Training (PTST) technique.

## B    FURTHER RESULTS

### B.1    DETAILS OF ANALYSIS OF ACCURACY-ONLY RLVR-TRAINED MLLMS

In this section, we present additional details and results to the analysis in Section 4.1.

We conduct our investigation from the following two perspectives: (1) The proportion of incorrect solving cases attributable to multimodal perception errors, and (2) a comparative analysis of the multimodal perception capabilities between the RLVR-trained MLLMs and their original counterparts. The former helps identify the bottleneck in the multimodal reasoning abilities of MLLMs, while the latter assesses whether their perception capabilities improve after accuracy-only RLVR training.

Specifically, we first train Qwen2-VL-7B-IT (Wang et al., 2024b) and Qwen2.5-VL-7B-IT (Bai et al., 2025) models on Geometry3K (Lu et al., 2021a) dataset using accuracy-only RLVR. We then

Table 5: Confusion matrix of Qwen2.5-VL-7B-IT evaluated on $\mathcal{D}_e$.

|  | Correct Answer | Wrong Answer |
| --- | --- | --- |
| Correct Perception | 19 | 3 |
| Wrong Perception | 11 | 17 |

Table 6: Confusion matrix of accuracy-only RLVR trained Qwen2.5-VL-7B-IT evaluated on $\mathcal{D}_e$.

|  | Correct Answer | Wrong Answer |
| --- | --- | --- |
| Correct Perception | 23 | 1 |
| Wrong Perception | 15 | 11 |

manually assess their CoT trajectories on the geometry reasoning subset of MathVista (Lu et al., 2024b), as well as the "Visual Dominant" and "Visual Only" subsets of MathVerse (Zhang et al., 2024b). These problems require both strong multimodal perception and logical reasoning capabilities, making them suitable for identifying potential weaknesses in MLLMs' reasoning performance. For each multimodal problem, we consider an MLLM to have made a perception error if its CoT trajectory contains an inaccurate visual description that is essential for reaching the correct final answer. All annotations are conducted by three well-trained annotators (all with at least a bachelor's degree). The template for human annotation is shown in Figure 4.

Let $\mathcal{D}_e$ denote the set of 50 problems randomly sampled from MathVista in Section 4.1. We provide additional results for Qwen2-VL-7B-IT in Tables 3 and 4, and for Qwen2.5-VL-7B-IT in Tables 5 and 6, where each problem is categorized based on the correctness of the model's final answer and visual perception.

Taking Qwen2.5-VL-7B-IT as an example, from the table, we observe that although the RLVR-trained model shows a significant improvement in problem-solving accuracy (from 30 to 38, i.e., 60% to 76%), its visual perception accuracy improves only marginally (from 22 to 24, i.e., 44% to 48%). Moreover, the proportion of problems with incorrect visual perception among those solved correctly even increases slightly (from 11/30 to 15/38, i.e., 36.7% to 39.4%), which also indicates that the multimodal perception capabilities of the RLVR-trained model have not been effectively improved.

### B.2 MANUAL EXAMINATION OF GENERATED VISUAL ANNOTATIONS

The correctness of the generated visual annotations is crucial for the effective application of the visual perception reward. To ensure their quality, we manually examined 100 randomly selected retrained samples to identify potential perception errors. Our analysis revealed that only 4 out of 100 samples contained minor perceptual inaccuracies, corresponding to a 96% perceptual accuracy in the visual annotations, which ensures an effective training of our Perception-R1.

The extracted visual annotations may contains implicitly reasoning process from the collected CoT trajectories, leading to implicit reasoning distillation. Although including implicit reasoning processes can also enhance the MLLM's logical reasoning capabilities through RL, this is inconsistent with our original intention of improving the MLLM's perceptual capabilities via the visual perception reward. To demonstrate that the performance gains of our Perception-R1 indeed stem from improved perception, we rigorously categorize the extracted visual annotations into "Visual Information", "Reasoning Information", and "Text Information". Here, "Visual Information" refers to information that can only be obtained from the image; "Reasoning Information" refers to information that does not directly exist in the image or problem text, but can be obtained through reasoning; and "Text Information" refers to information that can be obtained directly from the problem text. For comparison, we also conducted the same examination on the GeoQA+ dataset. We randomly examined 30 data instances sampled from the Geometry3K (Lu et al., 2021a) and GeoQA+ (Cao & Xiao, 2022) datasets, and we present the results in Table 7.

It is worth noting that any annotation that requires even mild reasoning to obtain is classified as "Reasoning Information" (e.g., Given an image of parallelogram ABCD, if the annotation states that "AB = CD", then it is classified as reasoning information). Most "Reasoning Information" are

---

**[Question ID]**
**[Image]**

**Question:** Chords $\overline{AC}$ and $\overline{DF}$ are equidistant from the center. If the radius of $\odot G$ is 26, find $DE$.

**Answer:** 24

**Response:** <think>To find the length of $DE$, we need to use the property that chords equidistant from the center of a circle are congruent. This means that $AC = DF$.

$\cdots\cdots$

Since $AC$ and $DF$ are equidistant and congruent in length, $DF = 48$. Applying the same logic to chord $\overline{DF}$, we use the right triangle $\triangle DGE$: $DG = \sqrt{RG^2 - GE^2} = \sqrt{26^2 - 10^2} = 24$. So, $DF = 2 \times DE$. Since $DF = 48$, we have: $DE = \frac{DF}{2} = \frac{48}{2} = 24$ </think><answer>$\boxed{24}$</answer>

**Please check whether there exist any multimodal perception error in Response that is critical for problem solving.**

**Judgment:** [Yes / No]

---

Figure 4: The template of human evaluation for multimodal perception.

Table 7: Classification results of extracted visual annotations from Geometry3K and GeoQA+.

| Dataset | All annotations | Visual Information | Reasoning Information | Text Information |
|---|---|---|---|---|
| Geometry3K | 99 | 81 (82%) | 10 (10%) | 8 (8%) |
| GeoQA+ | 114 | 29 (25%) | 41 (36%) | 44 (39%) |

such short statements rather than reasoning chains, minimizing the possibility of implicit reasoning distillation. Even with such a rigorous principle, the proportion of visual information in Geometry3K is still dominant (82%). In comparison, the proportion is only 25% for the GeoQA+ dataset. This is the core reason why we chose the Geometry3K dataset as our training data, as it provides better visual perception for our framework and isolates it from the influence of implicit reasoning.

### B.3 GENERALIZATION ANALYSIS OF PERCEPTION-R1

#### B.3.1 GENERALIZE TO QWEN2-VL MODEL

We apply our visual perception reward enhanced RLVR to train Qwen2-VL-7B-IT (Wang et al., 2024b) to demonstrate its generalizability and robustness. We present the experimental results in Table 8. Here, we compare against R1-VL (Zhang et al., 2025), as it is also trained from Qwen2-VL-7B-IT. From the table, we observe that Perception-R1-Qwen2 achieves the best performance on most benchmarks except MathVision (Wang et al., 2024a), demonstrating the effectiveness and generalizability of our method. Notably, similar to the full Perception-R1, Perception-R1-Qwen2 achieves a substantial improvement on the "Vision Only" subset of MathVerse (Zhang et al., 2024b) (39.2% vs. 30.1%), further validating the effectiveness of the proposed visual perception reward in enhancing the multimodal perception capabilities of MLLMs. We attribute the sub-optimal performance of Perception-R1-Qwen2 on MathVision to the limited diversity of the training dataset and believe this can be addressed by scaling up both the quantity and diversity of the training data.

#### B.3.2 GENERALIZE TO MULBERRY DATASET

To further demonstrate the generalizability of our method to other datasets, we conducted the same training pipeline on the data filtered from mulberry-260k (Yao et al., 2024), which contains 16.8K data and mainly from IconQA (Lu et al., 2021b), DVQA (Kafle et al., 2018) and does not contain any geometry data. During data collection stage, we employ Qwen3-VL-235B-A22B-Instruct (Team, 2025b) model to generate reasoning trajectories and employ Qwen3-Next-80B-A3B-Instruct (Team, 2025a) model to extract visual annotations because of their powerful multimodal reasoning and

Table 8: Experimental results of applying our method to Qwen2-VL-7B-IT. The best result is highlighted in red. † R1-VL-7B used WeMath and MathVision for training, their results on these benchmarks are omitted.

| Model | #Data | Math Benchmarks | | | | General Benchmarks | | | |
|---|---|---|---|---|---|---|---|---|---|
| | | MathVista testmini | MathVerse testmini | MathVision test | WeMath testmini | MMMU val | MMMU-Pro overall | MMStar val | EMMA full |
| Qwen2-VL-7B-IT | / | 58.6 | 31.1 | 16.7 | 42.3 | 46.9 | 29.6 | 56.0 | 24.5 |
| + GRPO | 1.4K | 64.5 | 38.1 | 19.7 | 54.6 | 51.4 | 32.4 | 56.3 | 24.3 |
| R1-VL-7B | 10K | 62.7 | 40.8 | -† | -† | 52.3 | 29.4 | 56.7 | 23.5 |
| **Perception-R1-Qwen2-7B** | **1.4K** | **64.9** | **42.3** | **20.4** | **60.0** | **53.1** | **35.2** | **56.9** | **25.1** |

Table 9: Experimental results of applying our method to our filtered mulberry dataset. The best result is highlighted in red.

| Model | #Data | Math Benchmarks | | | | General Benchmarks | | | |
|---|---|---|---|---|---|---|---|---|---|
| | | MathVista testmini | MathVerse testmini | MathVision test | WeMath testmini | MMMU val | MMMU-Pro overall | MMStar val | EMMA full |
| Qwen2.5-VL-7B-IT | / | 68.1 | 47.4 | 25.1 | 61.4 | 55.2 | 37.0 | 60.2 | 24.9 |
| + GRPO on Geometry3K | 1.4K | 73.3 | 51.3 | 26.6 | 69.5 | 58.0 | 38.2 | 63.1 | 24.9 |
| **Perception-R1-7B** | 1.4K | **74.2** | **54.3** | **28.6** | **72.0** | **60.8** | **42.4** | **64.5** | **27.5** |
| + GRPO on Mulberry | 16.8K | 72.6 | 46.2 | **27.8** | 66.8 | 52.1 | 42.0 | 62.1 | 26.4 |
| **Perception-R1-Mulberry-7B** | 16.8K | **73.4** | **51.2** | 27.1 | **69.9** | **59.1** | **42.2** | **62.6** | **27.2** |

language understanding capabilities. The prompts used in data collection and collection pipeline are same as those in Section C.2. We name the model trained on this dataset "Perception-R1-Mulberry-7B". The experimental results are present in Table 9.

From Table 9, we can observe that Perception-R1-Mulberry-7B still outperforms standard GRPO by 2.1 points on average across all benchmarks, demonstrating the effectiveness of our method. We believe the reason why Perception-R1-7B outperforms Perception-R1-Mulberry-7B is that the collected Mulberry data lacks math reasoning content (especially geometry) and mainly focuses on pure visual perception, which leads to worse performance on math benchmarks.

### B.4 ROBUSTNESS ANALYSIS OF PERCEPTION-R1

There are two factors can affect the robustness of Perception-R1: the correctness of visual perception reward and the factor $\gamma$ that controls the influence of visual perception reward to the final reward.

- Regarding the correctness of the visual perception reward, there are two types of factors that can impair it: the correctness of the extracted visual annotations and the correctness of the judgments produced by the judging LLM. We simulate these two types of noise by randomly flipping the judgments (i.e., $o_{i,j}$ in Eq. 4) produced by the Qwen2.5-32B-Instruct model from $1 \to 0$ or $0 \to 1$ at a fixed proportion. We conduct experiments with flipping proportions of 10% and 20%, and present the results in Table 10. From the table, we can observe that even with 20% of the visual perception reward corrupted, the model's average performance still surpasses that of GRPO, showcasing the robustness of our method. Notably, the performance degradation mainly comes from MathVista and MathVerse. This may be because these two benchmarks contain a large number of geometry test cases that are similar to our training data. In general benchmarks including MMMU and MMMU-Pro, the model trained with corrupted annotations still performs on par with Perception-R1-7B, further demonstrating the robustness of our training pipeline.

- Regarding the factor $\gamma$, we present in Table 11 the performance of models trained with different $\gamma$ values on each benchmark, as an extension of the average performance shown in Figure 3a. From the table, we observe that the average performances of models trained with different $\gamma$ values (except 0.0) are very similar, and all of them significantly surpass standard GRPO, demonstrating the robustness and effectiveness of our proposed method.

Table 10: Experimental results of randomly flipping judgment results $o_{i,j}$ at different proportions.

| Random flipping proportion | Math Benchmarks | | | | General Benchmarks | | | | Average |
|---|---|---|---|---|---|---|---|---|---|
| | MathVista testmini | MathVerse testmini | MathVision test | WeMath testmini | MMMU val | MMMU-Pro overall | MMStar val | EMMA full | |
| Qwen2.5-VL-7B-IT + GRPO | 73.3 | 51.3 | 26.6 | 69.5 | 58.0 | 38.2 | 63.1 | 24.9 | 50.6 |
| 0% (Perception-R1) | 74.2 | 54.3 | 28.6 | 72.0 | 60.8 | 42.4 | 64.5 | 27.5 | 53.0 |
| 10% | 72.2 | 51.1 | 29.1 | 69.2 | 60.5 | 42.9 | 63.1 | 28.3 | 51.9 |
| 20% | 70.0 | 50.6 | 27.4 | 70.7 | 60.9 | 42.0 | 62.1 | 27.9 | 51.5 |

Table 11: Experimental results of models trained with different $\gamma$ values.

| $\gamma$ values | Math Benchmarks | | | | General Benchmarks | | | | Average |
|---|---|---|---|---|---|---|---|---|---|
| | MathVista testmini | MathVerse testmini | MathVision test | WeMath testmini | MMMU val | MMMU-Pro overall | MMStar val | EMMA full | |
| 0.0 (GRPO) | 73.3 | 51.3 | 26.6 | 69.5 | 58.0 | 38.2 | 63.1 | 24.9 | 50.6 |
| 0.1 | 72.7 | 54.1 | 28.5 | 70.9 | 60.0 | 41.2 | 65.4 | 27.8 | 52.6 |
| 0.3 | 73.0 | 54.4 | 29.0 | 71.7 | 60.5 | 42.6 | 63.7 | 28.1 | 52.9 |
| 0.5 | 75.5 | 53.0 | 27.6 | 70.5 | 59.1 | 42.9 | 65.5 | 27.4 | 52.7 |
| 0.7 (Perception-R1) | 74.2 | 54.3 | 28.6 | 72.0 | 60.8 | 42.4 | 64.5 | 27.5 | 53.0 |
| 0.9 | 72.4 | 53.7 | 28.4 | 72.2 | 60.9 | 40.7 | 64.1 | 28.0 | 52.5 |

## B.5 Performance on Vision-Only Benchmarks

To further demonstrate the improved perception capabilities of Perception-R1 and Perception-R1-Qwen2 models, we compare their performance with baseline methods on Vision-Only subsets of MathVerse (Zhang et al., 2024b) and MMMU-Pro (Yue et al., 2024b) benchmarks in Table 12.

From the table, we can observe that both Perception-R1 and Perception-R1-Qwen2 surpass standard GRPO and previous SOTA method on these two vision-only benchmarks by a substantial margin. Specifically, Perception-R1 achieves an **average improvement of 2.6**, while Perception-R1-Qwen2 reaches an **average improvement of 6.3**. These results not only demonstrate that the multimodal perception capabilities of the Perception-R1 model series have been significantly enhanced but also validate the effectiveness of our proposed visual perception reward in boosting the multimodal perception capabilities of MLLMs.

## B.6 Analysis of Using Qwen2.5-VL-32B-IT as Reward Model

We provide the training dynamics of accuracy reward and visual perception reward of Perception-R1 and the variant using Qwen2.5-VL-32B-IT as the reward model in Figure 5. When using Qwen2.5-VL-32B-IT as the reward model, we provide it with both the image and the response generated by the policy model, and prompt it to output a consistency score in [0, 1], representing the degree of alignment between the image and the response. From Figure 5, we observe that the visual perception reward increases rapidly and saturates around 100 training steps. Meanwhile, the accuracy reward becomes consistently lower than that of Perception-R1 after the same point, indicating the presence of reward hacking when using Qwen2.5-VL-32B-IT as the reward model. This reward hacking issue undermines the multimodal reasoning performance of the resulting MLLM.

## B.7 Computational Costs Comparison

In this subsection, we compare the computational costs of Perception-R1 with representative baseline methods. We categorize the computational costs into **data preparation cost** and **training time cost**. For data preparation cost, we estimate it by counting the generated tokens in data curation process using Qwen2.5 Tokenizer. For training time cost, we calculate the total GPU-Hours used to train the model. We summarize the data preparation costs and training time costs of Perception-R1 and representative baseline methods in Table 13, with detailed explanations provided below:

**Data Preparation Cost**:

- **Perception-R1**: We collected CoT trajectories on 2,101 data samples (before filtering), resulting in a total of 1.01M tokens. For visual annotation extraction, we generated an additional 105K tokens. Thus, the total token cost is 1.1M tokens.

Table 12: Performance comparisons between Perception-R1 and baselines on vision-only subsets of MathVerse and MMMU-Pro. The best result is highlighted in red.

| Model | MathVerse vision-only | MMMU-Pro vision |
|---|---|---|
| *Qwen2.5-VL Models* | | |
| Qwen2.5-VL-7B-IT | 42.2 | 33.8 |
| + GRPO | 47.1 | 37.1 |
| R1-Onevision-7B | 41.9 | 30.7 |
| OpenVLThinker-7B | 39.5 | 35.3 |
| VLAA-Thinker-7B | 45.7 | 34.8 |
| SophiaVL-R1-7B | 43.3 | 37.6 |
| MM-Eureka-7B | 47.6 | 35.2 |
| Vision-R1-7B | 47.0 | 36.0 |
| **Perception-R1-7B** | **50.1** | **40.3** |
| Δ (Ours - Prev SOTA) | **+2.5** | **+2.7** |
| *Qwen2-VL Models* | | |
| Qwen2-VL-7B-IT | 30.1 | 26.6 |
| + GRPO | 32.4 | 29.8 |
| R1-VL-7B | 36.8 | 23.6 |
| **Perception-R1-Qwen2-7B** | **39.2** | **33.7** |
| Δ (Ours - Prev SOTA) | **+2.4** | **+10.1** |

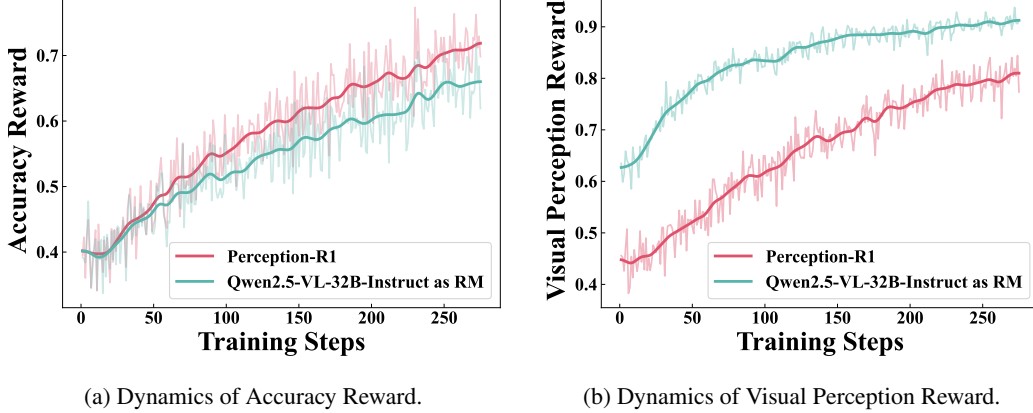

(a) Dynamics of Accuracy Reward.       (b) Dynamics of Visual Perception Reward.

Figure 5: Comparison of Accuracy and Visual Perception Rewards between Perception-R1 and the variant using Qwen2.5-VL-32B-IT as the Reward Model.

- **Vision-R1** prompted DeepSeek-R1 to produce 200K CoT trajectories. The total number of generated tokens is 134M.

- **MM-Eureka** performed pure RL on 15K self-collected samples without trajectory distillation, resulting in 0 token generation cost.

- **SophiaVL-R1** constructed the large-scale SophiaVL-R1-Thinking-156K dataset to train a thinking reward model for evaluating the thinking quality of the policy model during RL. This dataset was built by collecting CoT trajectories and leveraging powerful MLLM-based judgments, resulting in a total of 39.4M tokens.

- **VLAA-Thinker**: Although VLAA-Thinker did not perform SFT, its RL training dataset (VLAA-Thinking-Dataset) was selected and constructed by analyzing the captions and CoT trajectories generated by GPT-4o and DeepSeek-R1. Here we only count the tokens of RL dataset, which resulting in a total of 29.6M tokens.

Table 13: Computational costs comparisons between Perception-R1 and representative baselines, w.h.p. stands for "with high probability".

| Model | Data Preparation Cost (#Tokens) | Training Time Cost (GPU-Hours) |
|---|---|---|
| Perception-R1 | 1.1M Tokens | 167.4 A800-Hours (1.4K RL) |
| Vision-R1 | 134M Tokens | 3392 H800-Hours (200K SFT + 10K RL) |
| MM-Eureka | 0 | >167.4 A800-Hours w.h.p (15K RL) |
| SophiaVL-R1 | 34.9M Tokens | >167.4 A800-Hours w.h.p. (158K SFT + 130K RL) |
| VLAA-Thinker | 29.6M Tokens | <167.4 A800-Hours w.h.p. (25K RL) |
| OpenVLThinker | About 5.7M Tokens | >167.4 A800-Hours w.h.p. (25K SFT + RL) |
| R1-Onevision | >1.1M Tokens w.h.p | >167.4 A800-Hours w.h.p. (155K SFT + 10K RL) |
| R1-VL | 0 | >167.4 A800-Hours w.h.p. (260K SFT + 10K RL) |

- **OpenVLThinker** distilled 25K samples, of which only 3.2K (731K tokens) are publicly available. We estimate the total token count to be about 5.71M.

- **R1-OneVision** heavily relied on GPT-4o to enhance a subset of the LLaVA-OneVision dataset, but the augmented data is hard to separate from the original trajectories, making token cost estimation infeasible. Nonetheless, with 155K samples collected for SFT, its token generation cost likely exceeds 1.1M with high probability.

- **R1-VL** used the off-the-shelf mulberry-260K dataset for SFT, resulting in 0 token cost.

**Training Time Cost**: Since only Vision-R1 reported its detailed training setup, we can only provide a detailed comparison with it. For other baselines (R1-VL, R1-OneVision and OpenVLThinker), which require large-scale SFT, their training costs likely exceed that of Perception-R1 with high probability.

- **Perception-R1** can be trained in 16 hours using 16 A800 GPUs: 8 for serving the judging LLM and 8 for policy training. Each RL step takes an average of 154.3s, with judgment accounting for 47.5s, which means the 8 serving GPUs are idle 69.2% of the time and can be used for other API tasks. The total training cost of Perception-R1 is about 167.4 A800-Hours. Compared to standard GRPO, the only additional cost comes from judgment, increasing per-step time by 44.5%. However, due to the effectiveness of visual perception reward, Perception-R1 requires significantly less data than other baselines, ultimately resulting in a substantially lower total training cost.

- **Vision-R1** needed 32 H800 GPUs for about 10h SFT and 64 H800 GPUs for about 2 day RL, resulting in a total training cost of 3,392 H800-hours.

- **MM-Eureka** was trained on 15K distinct samples for 10 epochs, resulting in a total of 150K samples used for policy training. In contrast, Perception-R1 was trained on 1.4K distinct samples for 25 epochs, yielding 35K samples in total. Therefore, the training time cost of MM-Eureka is larger than that of Perception-R1 with high probability.

- **SophiaVL-R1** was trained on 130K RL samples, which required first training a thinking reward model on 158K SFT samples. During RL, SophiaVL-R1 needed incorporating the reward model into the RL training process. Therefore, the training time cost of SophiaVL-R1 is larger than that of Perception-R1 with high probability.

- **VLAA-Thinker** was trained on 25K distinct samples for a single epoch, resulting in 25K samples in total, which is fewer than those used for training Perception-R1. Consequently, the training time cost of VLAA-Thinker is likely lower than that of Perception-R1.

## B.8 CASE STUDIES

We provide qualitative comparisons as follows, where cases 1-3 are from MathVista (Lu et al., 2024b), case 4 is from MMMU (Yue et al., 2024a) and case 5 is from MMStar (Chen et al., 2024c). These cases reveal that MLLMs trained with accuracy-only RLVR often exhibit similar perception errors as their original counterparts, which in turn lead to incorrect problem solving. In contrast, our proposed Perception-R1 effectively alleviates such perception failures, enabling the model to better understand the visual content and ultimately arrive at the correct solution. Notably, Perception-R1 demonstrates a distinct "describe-then-solve" behavior compared to others, where it explicitly interprets visual

content prior to reasoning. This leads to more accurate solutions in multimodal reasoning and further validates the effectiveness of the proposed visual perception reward.

---

**Case Study 1 (MathVista)**

**Question:** Find $y$.

**Choices:** A. $16\sqrt{2}$, B. $16\sqrt{3}$, C. 32, D. $16\sqrt{5}$

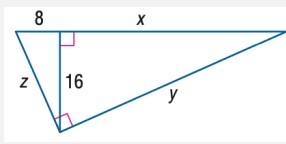

---

**Qwen2.5-VL-7B-IT:**
To find the length of $y$, we need to use the Pythagorean theorem in the context of the right triangles formed in the diagram.
First, let's identify the right triangles: 1. The triangle with legs of 8 and 16. 2. The triangle with legs of $x$ and 16.
The hypotenuse of the first triangle is $y$, and the hypotenuse of the second triangle is $z$.
Using the Pythagorean theorem for the first triangle: $y^2 = 8^2 + 16^2$, $y = \sqrt{320} = 8\sqrt{5}$
So, the correct answer is: (D) $16\sqrt{5}$

---

**Accuracy-only RLVR trained Qwen2.5-VL-7B-IT:**
<**think**> To find the length $y$, we need to recognize that the two triangles are similar by AA (Angle-Angle) similarity postulate, as both have a right angle and share another angle. This means the ratios of corresponding sides will be equal. We can set up the proportion based on the given sides: $\frac{x}{y} = \frac{8}{16}$.
However, we need to find $y$ directly. Notice that the smaller triangle is a 45-45-90 triangle scaled up, which means the hypotenuse $y$ can be found using the relationship in a 45-45-90 triangle where the hypotenuse is $\sqrt{2}$ times the leg. But here, we can use the Pythagorean theorem in the larger triangle formed by the height and the base: $y = \sqrt{16^2 + 8^2} = \sqrt{256 + 64} = \sqrt{320} = \sqrt{64 \times 5} = 8\sqrt{5}$
But upon rechecking with the similar triangle ratio directly: $y = 16\sqrt{2}$
But let's re-evaluate with the direct similar triangle ratio: $\frac{y}{16} = \frac{8}{8} \Rightarrow y = 16\sqrt{2}$
But upon rechecking the options and direct similar triangle ratio: $y = 32$
But let's recheck with Pythagorean in larger triangle: $y = \sqrt{16^2 + 8^2} = \sqrt{256 + 64} = \sqrt{320} = 16\sqrt{5}$ </**think**>
<**answer**> $16\sqrt{5}$ </**anwer**>

---

**Perception-R1:**
<**think**> Looking at the image, we can see a right triangle divided into two smaller right triangles by the altitude (16) drawn to the hypotenuse (x). According to the geometric mean theorem (also known as the altitude-on-hypotenuse theorem), the altitude (16) squared is equal to the product of the two segments of the hypotenuse it creates, which are 8 and x. So, we have: $16^2 = 8 \cdot x$, $x = 32$.
Now, we need to find y, the hypotenuse of the larger right triangle. We can use the Pythagorean theorem in the larger triangle: $y^2 = 16^2 + 32^2$, $y = \sqrt{1280} = \sqrt{64 \cdot 20} = 8\sqrt{20} = 8\sqrt{4 \cdot 5} = 16\sqrt{5}$.
So, the answer is $16\sqrt{5}$. </**think**>

<**answer**> $16\sqrt{5}$ </**answer**>

---

---

**Case Study 2 (MathVista)**

**Question:** Find $x$.
**Choices:** A. 6.00, $\boxed{\text{B. 9.45}}$, C. 18.9, D. 37.8

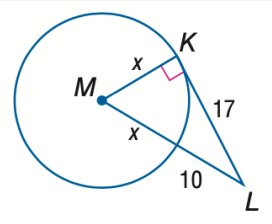

---

**Qwen2.5-VL-7B-IT:**
To solve for the value of $x$ in the given figure, we need to use the properties of tangents and the Pythagorean theorem.
1. **Identify the given information:**
$KL$ is tangent to $\odot M$ at point $K$, $MK$ is the radius of the circle, $MK = x$, $KL = 17$, $ML = 10$.
2. **Use the Pythagorean theorem:**
Since $KL$ is tangent to the circle at $K$, $MK$ is perpendicular to $KL$. Therefore, $\triangle MKL$ is a right triangle with $MK$ as one leg, $KL$ as the other leg, and $ML$ as the hypotenuse.
3. **Apply the Pythagorean theorem:**

$$ML^2 = MK^2 + KL^2$$

Substitute the known values:
$$100 = x^2 + 289$$

4. **Solve for $x^2$:**

$$x^2 = 100 - 289$$
$$x^2 = -189$$

5. **Check the calculation:**
There seems to be an error in the setup or the given values. Let's recheck the problem statement and the reference solution.
Given the reference solution, the correct value of $x$ is 9.45.
Thus, the correct answer is: $\boxed{B}$

---

**Accuracy-only RLVR trained Qwen2.5-VL-7B-IT:**
<think> To find the value of $x$, we need to use the property that the radius drawn to the point of tangency is perpendicular to the tangent line. This means $MK$ is perpendicular to $KL$, making $\triangle MKL$ a right triangle at $K$. We can use the Pythagorean theorem in $\triangle MKL$:
$ML^2 = MK^2 + KL^2$
Given $ML = 17$, $MK = x$, and $KL = 10$, we substitute these values into the equation:
$17^2 = x^2 + 10^2$, $x = \sqrt{189} = \sqrt{9 \times 21} = 3\sqrt{21}$.
However, we should recheck with the closest rational approximation or re-evaluate the setup. Let's re-use the Pythagorean directly:

$$x = \sqrt{17^2 - 10^2} = \sqrt{289 - 100} = \sqrt{189} \approx 13.75$$

But let's recheck with options:

$$x = 9.45 \text{ seems close checking } 9.45^2 + 10^2 = 89.025 + 100 = 189$$

So, $x \approx 9.45$
</think>

**<answer>** $\boxed{B}$ **</answer>**

---

**Perception-R1:**
**<think>** Looking at the image, we can use the tangent-tangent theorem which states that if two tangents are drawn to a circle from an external point, they are equal in length. However, in this case, we are dealing with a tangent from an external point $L$ to the point of tangency $K$ and the radius $MK$ which is perpendicular to the tangent at the point of tangency. This means $MK \perp KL$, and by the Pythagorean theorem in $\triangle MKL$, we have:

$$MK^2 + KL^2 = ML^2.$$

Here, $MK = x$, $KL = 17$, and $ML = 10 + x$ (since $ML = MK + KL$ and $MK = x$). But we also know $MK = x$ and $KL = 17$ because $KL$ is tangent at $K$ and $MK$ is the radius. Using the Pythagorean theorem: $x^2 + 17^2 = (10 + x)^2$. Expanding and simplifying: $x^2 + 289 = 100 + 20x + x^2$, $189 = 20x$, $x = \frac{189}{20} = 9.45$. So the answer is $\boxed{B}$. **</think>**
**<answer>** $\boxed{B}$ **</answer>**

---

## Case Study 3 (MathVista)

**Question:** How many baseballs are there?.
**Answer:** 20

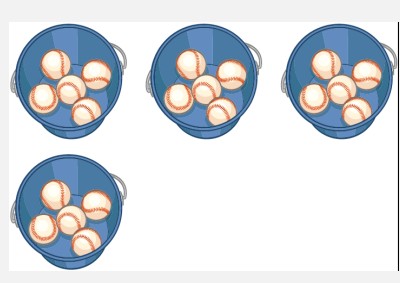

---

**Qwen2.5-VL-7B-IT:**
To determine the total number of baseballs, we need to count the number of baseballs in each bucket and then sum them up.

1. The first bucket contains 6 baseballs.
2. The second bucket contains 6 baseballs.
3. The third bucket contains 6 baseballs.
4. The fourth bucket contains 6 baseballs.

Adding these together:
6 (from the first bucket) + 6 (from the second bucket) + 6 (from the third bucket) + 6 (from the fourth bucket) = 24
So, the total number of baseballs is 24.

---

**Accuracy-only RLVR trained Qwen2.5-VL-7B-IT:**
**<think>** To determine the total number of baseballs, we need to count the baseballs in each bucket and then sum them up.

- The first bucket has 6 baseballs.
- The second bucket has 6 baseballs.
- The third bucket has 6 baseballs.
- The fourth bucket has 6 baseballs.

Adding them together: $6 + 6 + 6 + 6 = 24$. **</think>**
**<answer>** $\boxed{24}$ **</answer>**

**Perception-R1:**
<think> Looking at the image, I can see four buckets, and each bucket contains 5 baseballs. So, I multiply the number of buckets by the number of baseballs per bucket: 4 buckets * 5 baseballs/bucket = 20 baseballs.
Therefore, the answer is 20 . </think>
<answer> 20 </answer>

## Case Study 4 (MMMU)

**Question:** Which arrow in <image 1>points to a large ribosomal unit?
**Answer:** $I$

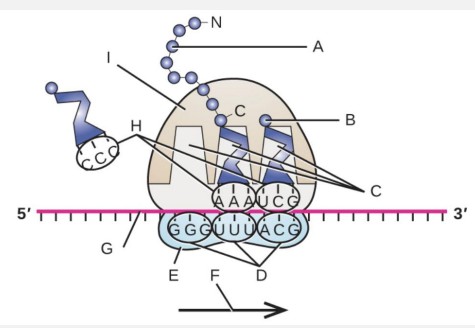

**MM-Eureka-Qwen-7B:**
<think> The image shows a diagram of a ribosome, which is a complex structure involved in protein synthesis. The large ribosomal unit is the one that is larger in size and is involved in the initiation of translation. In the diagram, the large ribosomal unit is the one that is not part of the small subunit and is typically associated with the mRNA and tRNA. The arrow in the diagram that points to the large ribosomal unit is the one that is not part of the small subunit and is associated with the mRNA and tRNA. In the diagram, the large ribosomal unit is the one that is not part of the small subunit and is associated with the mRNA and tRNA. The arrow that points to the large ribosomal unit is the one that is not part of the small subunit and is associated with the mRNA and tRNA. Therefore, the correct answer is the one that is not part of the small subunit and is associated with the mRNA and tRNA.
The correct answer is: G.</think>
<answer> The answer is G </answer>

**Vision-R1-7B:**
<think> Okay, let's see. The question is asking which arrow points to a large ribosomal unit. The options are A through I. From the image provided, it displays that arrow H points to a large ribosomal unit. The image also displays that the large ribosomal unit is the larger of the two subunits. So, according to the given information, the correct answer should be H. The other options like A, B, C, etc., are labeled with different parts of the ribosome or mRNA, but the question specifically asks for the large ribosomal unit, which is labeled by H. </think>
<answer> Final Answer: H </answer>

**Perception-R1:**
<think> Looking at the image, we need to identify which arrow points to a large ribosomal unit. The large ribosomal unit is typically the structure that includes both the small and large subunits, which are often shown as a single unit in diagrams of translation.

- A, B, and C appear to be parts of the ribosome, but not the entire unit.
- D, E, and F seem to be components of the mRNA and tRNA, not the ribosome.
- G looks like it might be pointing to the mRNA strand.
- H appears to be pointing to the tRNA.

- I looks like it might be pointing to the large ribosomal subunit, as it is positioned above the mRNA and tRNA, which is consistent with the large ribosomal unit in a ribosome.

Given the options, the arrow that points to the large ribosomal unit is $\boxed{I}$.
**</think>**
**<answer>** $\boxed{I}$ **</answer>**

---

**Case Study 5 (MMStar)**

**Question:** Which corner doesn't have any plates?
**Choices:** A: top-right, B: top-left, C: bottom-left, D: bottom-right
**Answer:** $D$

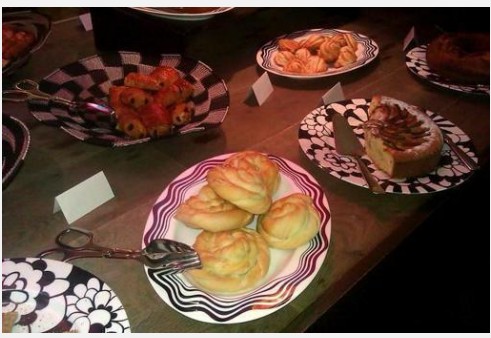

**MM-Eureka-Qwen-7B:**
**<think>** The image shows a table with various plates of food. By observing the arrangement, we can see that the top-right corner of the image is empty, as there are no plates in that area. The other corners have plates with food on them. Therefore, the corner that doesn't have any plates is the top-right corner.
The answer is: A: top-right. **</think>**
**<answer>** The answer is $\boxed{A}$ **</answer>**

---

**Vision-R1-7B:**
**<think>** Okay, let's see. The question is asking which corner doesn't have any plates. The options are top-right, top-left, bottom-left, and bottom-right. From the image provided, it displays that the top-right corner is clearly visible and does not have any plates. The other corners—top-left, bottom-left, and bottom-right—do have plates. So the answer should be the top-right corner, which is option A. **</think>**
**<answer>** Final Answer: $\boxed{A}$ **</answer>**

---

**Perception-R1:**
**<think>** Looking at the image, I can see plates with various baked goods in the top-left, top-right, and bottom-left corners. The bottom-right corner appears to be empty, without any plates or food items. Therefore, the corner that doesn't have any plates is the bottom-right corner. So, the answer is $\boxed{D}$. **</think>**
**<answer>** $\boxed{D}$ **</answer>**

## C    IMPLEMENTATION DETAILS

### C.1    EXPERIMENTAL SETTINGS

In the experiments, we adopt Qwen2.5-VL-7B-IT (Bai et al., 2025) as our base model and train it using the EasyR1 (Zheng et al., 2025) framework. During training, we adopt Qwen2.5-32B-IT (Yang et al., 2024) as the judging LLM and set the learning rate to 1e-6 with a warmup ratio of 0.05. The base model is trained for a total of 25 epochs with a global batch size of 128. Following

---

**Prompt for RLVR Training**

You FIRST think about the reasoning process as an internal monologue and then provide the final answer. The reasoning process MUST BE enclosed within <think></think>tags, and the answer process MUST BE enclosed within <answer></answer>tags. The final answer MUST BE put in

boxed in <answer></answer>tags.

---

Figure 6: Prompt used for all RLVR training experiments in this work.

previous works (Meng et al., 2025; Yu et al., 2025), we remove the KL penalty from Eq.3 during RL training to achieve better performance, i.e., $\delta = 0$. Additionally, the coefficients in Eq.5 are set to $\alpha = 0.1, \beta = 0.9$, and $\gamma = 0.7$, where $\alpha$ and $\beta$ follow the settings in the EasyR1 (Zheng et al., 2025) codebase. The training process takes about 16 hours on 16 NVIDIA-A800-80G GPUs.

## C.2 PROMPTS

In this subsection, we provide the prompts that used for RLVR training (Figure 6), the prompt for extracting visual annotations from CoT trajectories (Figure 7), and for judging consistency between visual annotations and rollouts generated by policy models (Figure 8).

## D BROADER IMPACTS

In this paper, we propose Perception-R1 by introducing a novel visual perception reward to enhance the multimodal perception and reasoning capabilities of MLLMs. Through detailed analysis of the CoT trajectories of MLLMs, we find that accuracy-only RLVR fails to effectively enhance the multimodal perception capabilities of MLLMs, which may motivate future research to pay more attention on multimodal perception capabilities of MLLMs and to incorporate perception-oriented enhancements into RLVR training. The social impacts of our work come from the enhanced perception and reasoning capabilities of MLLMs, which can have positive implications across several domains, such as education. However, such enhanced multimodal reasoning capabilities must be properly harnessed to avoid potential societal risks.

## E THE USAGE OF LLMS STATEMENTS

In addition to positioning MLLMs as the primary focus of our research and experiments, this paper also leverages LLMs/MLLMs in three key areas: data collection, core implementation of the proposed method, and manuscript refinement. During data collection stage, we employed Gemini-2.5-Pro (Team et al., 2023) to generate detailed CoT trajectories for the Geometry3K (Lu et al., 2021a) dataset, which were then used by a separate text-only LLM to extract visual annotations $\mathcal{V}$. During model training, a text-only LLM was also introduced to assess the consistency between the policy model's responses and the extracted visual annotations $\mathcal{V}$, forming the basis of the visual perception reward. For manuscript writing, we utilized GPT-4o to assist with language polishing.

---

**Prompt for Extracting Visual Annotation**

Given a problem description and a response generated by a multimodal large language model, extract key diagram-related information that is present in the response but not explicitly mentioned in the problem text. Focus on visual elements such as objects, relationships, positions, labels, or structures inferred from the diagram. Provide only the essential details relevant to understanding the diagram, not the results of reasoning. **Make sure the visual key information is written in English.**

Here are some examples:
**Example 1**:
**Problem**: Find the measure of $\angle 7$ if $\overline{AB} \perp \overline{BC}$.
**Response**: Let E be the vertex on the horizontal line where angles 4, 7, and the $40°$ angle meet. The angle labeled $40°$ and angle 4 are vertically opposite angles. Vertically opposite angles are equal. Therefore, the measure of angle 4 is $40°$.

$$\angle 4 = 40°$$

Angles 4 and 7 form a linear pair on the straight horizontal line. Angles in a linear pair are supplementary, meaning their sum is $180°$. Substitute the value of $\angle 4$ into the equation:

$$40° + \angle 7 = 180°$$

Subtract $40°$ from both sides to find the measure of angle 7:

$$\angle 7 = 180° - 40°$$

$$\angle 7 = 140°$$

The condition $\overline{AB} \perp \overline{BC}$ means that the angle $\angle ABC = 90°$. The right angle symbol at vertex B indicates that $\angle 5 + \angle 6 = 90°$. This information is not needed to find the measure of $\angle 7$. Final Answer: The final answer is $\boxed{140°}$
**Visual Key Information**:
<info1>The angle labeled $40°$ and angle 4 are vertically opposite angles.</info1>
<info2>Angles 4 and 7 form a linear pair on the straight horizontal line.</info2>
**Example [2-3]**:
. . . . . .

**Problem**: {Problem text}
**Response**: {Response from powerful MLLMs}
**Visual Key Information**:

---

Figure 7: Prompt used to extract visual annotations $\mathcal{V}$ from CoT trajectories. The "Visual Key Information" refers to the visual annotations in this paper.

---

**Prompt for Judging LLM $\Phi$**

Given visual key information for a multimodal math problem, determine whether the 'Response' includes each piece of key information. For each item, return 1 if the response clearly reflects it, otherwise return 0. Respond using the format: <info1>1 or 0</info1>, <info2>1 or 0</info2>, etc. Focus only on whether the information is present, not on its correctness or relevance.

Here are some examples:
**Example 1**:
**Visual Key Information**:
<info1>$JKLM$ is a parallelogram.</info1>
<info2>Length of side $JK$ is given as $3f - 6$.</info2>
<info3>Length of opposite side $ML$ is given as $2f + 8$.</info3>
**Response**: To find the value of $f$ in the parallelogram, we need to use the properties of a parallelogram. Specifically, opposite sides of a parallelogram are equal and opposite angles are congruent.
Given:

- $\angle J$ is $56°$
- $\angle M$ is $(3d - 2)°$
- $\overline{JK}$ is $3f - 6$, $\overline{ML}$ is $2f + 8$

Since $\overline{JK}$ and $\overline{ML}$ are opposite sides of the parallelogram, we have:

$$3f - 6 = 2f + 8$$

Next, we solve for $f$:
Subtract $2f$ from both sides:
$$f - 6 = 8$$

Add 6 to both sides:
$$f = 8 + 6$$
$$f = 14$$

Thus, the value of $f$ is $\boxed{14}$.
**Judgment**: <info1>0</info1><info2>1</info2><info3>1</info3>
**Example [2-3]**:
......

**Visual Key Information**: {Extracted visual annotations}
**Response**: {Rollout of policy model $\pi_\theta$}
**Judgment**:

Figure 8: Prompt used to judge consistency between visual annotations $\mathcal{V}$ and rollouts during training. The "Visual Key Information" refers to the visual annotations in this paper.

