# OpenReview forum: "Perception-R1: Advancing Multimodal Reasoning Capabilities of MLLMs via Visual Perception Reward"
_ICLR.cc/2026/Conference — ICLR 2026 Poster_

### Official Review · Reviewer_keG9 · 2025-10-27

**Soundness:** 3
**Presentation:** 3
**Contribution:** 2
**Rating:** 6
**Confidence:** 1

**Summary:**

This paper proposes Perception-R1, a method to enhance multimodal reasoning in Multimodal Large Language Models (MLLMs) by introducing a visual perception reward alongside the standard accuracy reward in Reinforcement Learning with Verifiable Rewards (RLVR).

Key Contributions:

1. Problem Identification: Through McNemar's test, the authors demonstrate that existing accuracy-only RLVR fails to improve MLLMs' multimodal perception capabilities, which they identify as a major bottleneck.
2. Method: They introduce a visual perception reward that extracts textual visual annotations from CoT trajectories and uses a judging LLM to assess consistency between these annotations and model responses, which provides additional training signal beyond answer correctness.
3. Results: Using only 1,442 training samples, Perception-R1 achieves SOTA performance across multiple benchmarks, outperforming Vision-R1 (which uses 200K samples) and other baselines.

**Strengths:**

1. The methods introduced in the paper provide denser reward for the reinforcement learning process, which accurately solve the problem identified by the authors.

2. The experiments are extensive and thorough.

3. Using only 1,442 training samples, Perception-R1 achieves SOTA performance across multiple benchmarks, outperforming Vision-R1 (which uses 200K samples) and other baselines.

**Weaknesses:**

Concern 1: Marginal Improvement Beyond GRPO Baseline. According to Table 2, the performance improvements appear to be primarily driven by GRPO rather than the proposed visual perception reward. The reviewer notes that GRPO is also used in Vision-R1, making it unclear how much of the improvement is attributable to the novel contribution versus the baseline RL algorithm.

Concern 2: Judging LLM Quality Dependency. Figure 3b shows that when using smaller judging LLMs (e.g., Qwen2.5-7B or 14B), the performance sometimes drops below even the base model performance (e.g., MathVerse: 46.1% vs 47.4% baseline; MathVision: 24.2% vs 25.1% baseline). This raises questions about the robustness and practical applicability of the method when high-quality judging models are unavailable.

**Questions:**

The reviewer suggests that the author could conduct additional experiments where:

(1) Vision-R1 trained on the same 1,442 samples used in this paper.

(2) Using the same model to generate CoT trajectories for fair comparison.

---

> ### Author Response · Authors · 2025-11-22
> **[Part 1/1] Author Response**
>
> Thank you for taking the time to review our work. We address your questions as follows:
>
> ---
>
> **W1 & Q1**: Marginal Improvement Beyond GRPO Baseline. How much of the improvement is attributable to the novel contribution versus the baseline RL algorithm?
>
> **A**: We would like to clarify that the improvements are not marginal. **Both of our Perception-R1 and Perception-R1-Qwen2 surpass standard GRPO on every benchmark, outperforming it by 2.4 and 2.0 average points across 8 benchmarks, respectively. Moreover, on the vision-only subsets of MathVerse and MMMU-Pro, the performance gains are even more substantial, exceeding GRPO by 3.1 and 5.4 average points, respectively.** We further conduct McNemar’s test between the GRPO-trained MLLM and Perception-R1 (Section 4.1 and Lines 403–407), which **confirms that Perception-R1 exhibits significantly stronger multimodal perception capabilities, while the GRPO-trained MLLM does not.** These results demonstrate that our method provides meaningful and consistent improvements over standard GRPO.
>
> As you suggested, we use the same 1,442 geometry3k data and same training settings to train Vision-R1-CI-7B model (cold initialization of Vision-R1) using GRPO, to eliminate the influence of different RL data compared to Vision-R1. Since Vision-R1-CI was trained on some MathVision and WeMath data, we exclude them from comparison. The experimental results are as follows:
>
>
> |Model Name|MathVista|MathVerse|MMMU|MMMU-Pro|MMStar|EMMA|Avg|
> |:-|:-:|:-:|:-:|:-:|:-:|:-:|:-:|
> |Qwen2.5-VL-7B-Instruct|68.1|47.4|55.2|36.4|60.2|24.9|48.7|
> |Vision-R1-CI + GRPO on Geometry3K|69.9|46.0|55.3|37.9|63.8|18.1|48.5|
> |**Perception-R1-7B**|**74.2**|**54.3**|**60.8**|**42.4**|**64.5**|**27.5**|**54.0**|
>
> From the table, we observe that the Vision-R1-CI model trained on our small-scale dataset with GRPO performs even worse than the base model on average. **This finding shows that the superiority of Perception-R1 over Vision-R1 does not come from the training data but from the effectiveness of our proposed visual perception reward and training pipeline, even though it uses less than 1/20 of the GPU hours required by Vision-R1 for RL training** (see Appendix B.6).
>
> ---
>
> **W2**: The robustness and practical applicability of the method when high-quality judging models are unavailable.
>
> **A**: To test the robustness of the method, we randomly flipped the Qwen2.5-32B-IT generated judgment with different ratios to simulate the erroneous judgments occured in weaker judging LLMs. We conduct experiments with 10% and 20% random flipped judgments, and the evaluation results are as follows:
>
> |Random flip ratio|MathVista|MathVerse|MathVision|WeMath|MMMU|MMMU-Pro|MMStar|EMMA|Avg|
> |:-|:-:|:-:|:-:|:-:|:-:|:-:|:-:|:-:|:-:|
> |Qwen2.5-VL-7B-Instruct + GRPO|73.3|51.3|26.6|69.5|58.0|38.2|63.1|24.9|50.6|
> |**0% (Perception-R1-7B)**|**74.2**|**54.3**|28.6|**72.0**|**60.8**|42.4|**64.5**|27.5|**53.0**|
> |10%|72.2|51.1|**29.1**|69.2|60.5|**42.9**|63.1|**28.3**|51.9|
> |20%|70.0|50.6|27.4|70.7|60.9|42.0|62.1|27.9|51.5|
>
> We can see that even with 20% erroneous judgments, the average performance of the trained model still outperforms standard GRPO by 0.9 on average across all benchmarks, demonstrating the robustness of our method.
>
> Regarding the concern about the lack of high-quality judging models, we suspect that your concern relates to the additional computational overhead introduced by our training pipeline, since the open-source models themselves are always there. We believe this issue will be gradually addressed with the foundation models continue to advance. On the one hand, the judging model is only required to determine whether a short piece of text (the visual annotation) appears in a longer piece of text (the policy model’s rollouts). **This simplifed task relies solely on the basic language understanding ability, without requiring any data-related knowledge or high-level reasoning capabilities. This suggests that the ability required to make a correct judgment for judging LLM does not increase with the amount of knowledge or reasoning involved in the data itself.** On the other hand, as foundation models continue to advance, their fundamental language understanding capabilities also improve. In our current pipeline, adopting the Qwen2.5-32B-IT model introduces 30.8% more computational overhead compared with standard GRPO. However, **we believe that this overhead can be substantially reduced by employing a smaller model that retains language understanding capabilities comparable to those of Qwen2.5-32B-IT.**
>
> ---
>
> **Q2**: Using the same model to generate CoT trajectories for fair comparison.
>
> **A**: We apologize for not fully understanding the experiment you mentioned. It would be helpful if you could further clarify **which model should be used to generate the CoT trajectories** and **which model should be trained using these trajectories**, so that we can ensure a fair comparison and conduct a more comprehensive experiment.

---

### Official Review · Reviewer_5AFL · 2025-10-31

**Soundness:** 3
**Presentation:** 3
**Contribution:** 3
**Rating:** 8
**Confidence:** 4

**Summary:**

This paper introduces Perception-R1, a method to improve multimodal reasoning by fixing the key bottleneck of poor visual perception. The authors find that standard accuracy-only reinforcement learning (RLVR) fails to correct these perception errors, as models can guess the right answer despite flawed visual understanding. Perception-R1 addresses this by adding a visual perception reward. This reward is calculated by a judging LLM that compares the model's response to pre-extracted "visual annotations" (key visual facts) from correct solutions. Experiments show this method achieves superior performance on multiple benchmarks with high data efficiency, using only 1,442 training samples.

**Strengths:**

1. The paper provides a clear and compelling statistical analysis (using McNemar's test) of accuracy-only RLVR-trained MLLMs. This builds a strong case that a significant bottleneck for current models is indeed multimodal perception, not just high-level reasoning.

2. The proposed visual perception reward is intuitive and cleverly designed. By having an LLM judge responses against verifiable, extracted annotations rather than training a holistic reward model, the method directly targets the identified bottleneck. This approach appears significantly more robust to the reward hacking that can harm end-to-end MLLM-as-reward-model RLVR.

3. The performance gains achieved using only 1,442 training samples are impressive. This strongly suggests that a higher-quality, more targeted reward signal (i.e., combining perception and accuracy) can be far more sample-efficient than simply scaling up data for a sparser, accuracy-only reward.

4. The method delivers substantial performance improvements not only on its training domain (math/geometry) but also, surprisingly, across several general-domain benchmarks, outperforming baselines that used orders of magnitude more data.

**Weaknesses:**

1. Limited analysis of generalization: The model's strong generalization from geometry-only training (Geometry3K) to general-domain benchmarks (like MMMU and MMStar) is a key result, but it is not fully explained. The authors hypothesize that they are improving a foundational perception capability, but the link between 'perceiving geometry diagrams' and 'perceiving real-world images' could be strengthened. To make this claim more concrete, the authors could include:

- Qualitative analysis on general benchmarks: Provide qualitative examples from MMMU or MMStar. Does the Perception-R1 model now exhibit the same "describe-then-solve" behavior on these general-domain images? Where do the baseline models fail on perception in these tasks? Is Perception-R1 delivering more accurate visual perception in these examples?

- Error breakdown on general benchmarks: Conduct a small-scale error analysis on a subset of a general benchmark (like MMMU-Pro, where they show strong results). What percentage of the baseline's failures on these tasks are due to perception errors, and what percentage of those specific errors does Perception-R1 fix? This would directly support the claim of foundational perception improvement.

2. Dependence on a single training data domain: The reliance on Geometry3K, while clearly effective, is a potential limitation. The data curation pipeline itself seems general, but its effectiveness has only been demonstrated on this one domain. An ablation study training on a different domain (e.g., general textbook diagrams, or even a VQA dataset) using the same pipeline would be highly valuable. This would help demonstrate the general applicability of the Perception-R1 framework, distinguishing its contribution from the (clearly very effective) choice of geometry data as a training source.

**Questions:**

Please see the weaknesses above.

---

> ### Author Response · Authors · 2025-11-22
> **[Part 1/2] Author Response**
>
> Thank you for your affirmation of our motivation and constructive suggestions. For your concerns:
>
> ---
>
> **W1**: Provide qualitative examples from MMMU or MMStar. Does the Perception-R1 model now exhibit the same "describe-then-solve" behavior on these general-domain images? Where do the baseline models fail on perception in these tasks? Is Perception-R1 delivering more accurate visual perception in these examples?
>
> **A**: Thank you for your suggestions. We will add more cases from MMMU and MMStar in Appendix in the revised version of our paper.
>
> > Does the Perception-R1 model now exhibit the same "describe-then-solve" behavior on these general-domain images?
>
> Yes, in our case studies, most of the solving cases follow the "describe-then-solve" behavior even on general benchmarks. It is worth noting that we do not enforce this behavior during RL training. We only encourage image descriptions in the solving trajectories, and the model itself develops and follows such a solving pattern.
>
> > Where do the baseline models fail on perception in these tasks? Is Perception-R1 delivering more accurate visual perception in these examples?
>
> In the cases we provide from MathVista (see Appendix B.7), Perception-R1 indeed delivers more accurate visual perception compared to standard GRPO and the base model (e.g., correctly identifying 5 balls in each bin in Case 3 but other models do not). We will include additional comparisons with baselines, including Vision-R1 and MM-Eureka, on general benchmarks such as MMMU and MMStar in Appendix B.7 in the revised version of our paper.
>
> ---
>
> **W2**: Conduct a small-scale error analysis on a subset of a general benchmark (e.g., MMMU-Pro). What percentage of the baseline's failures on these tasks are due to perception errors, and what percentage of those specific errors does Perception-R1 fix?
>
> **A**: We conducted case studies on 40 randomly sample problems from MMMU-Pro, and manually classified them into 2 classes: final answer correct or not, and visual perception correct or not. We present the results on Perception-R1, Vision-R1 and MM-Eureka as follows:
>
> |Perception-R1|Correct Answer|Wrong Answer|
> |:-|:-:|:-:|
> |Correct Perception|19|11|
> |Wrong Perception|0|10|
>
> |Vision-R1|Correct Answer|Wrong Answer|
> |:-|:-:|:-:|
> |Correct Perception|15|9|
> |Wrong Perception|3|13|
>
> |MM-Eureka|Correct Answer|Wrong Answer|
> |:-|:-:|:-:|
> |Correct Perception|12|12|
> |Wrong Perception|2|14|
>
> For the percentage of each model's failures on MMMU-Pro that are due to perception errors, the statistics are 47.6%, 61.9%, and 53.8%. Among all problems, Perception-R1 achieves a correct perception rate of 75%, while the corresponding rates for Vision-R1 and MM-Eureka are both 60%. From these results, although Perception-R1 improves perceptual capabilities compared to the baseline models, perceptual errors remain a bottleneck that limits problem-solving accuracy on MMMU-Pro, and further improvements are still needed in the future. **Among all 40 problems, Perception-R1 successfully fixes 6 perceptual errors detected in Vision-R1 and 6 in MM-Eureka, corresponding to proportions of 37.5% and 37.5%, respectively.**

---

> ### Author Response · Authors · 2025-11-22
> **[Part 2/2] Author Response**
>
> **W3**: Ablation study training on a different domain using the same pipeline.
>
> **A**: Thank you for your insightful suggestion! To isolate the contribution from the geometry data, we collect a more perception-dedicated training data from Mulberry [1], which contains 16.8K data and mainly from IconQA, DVQA and do not contain any geometry data. During data collection stage, we employ Qwen3-VL-235B-A22B-Instruct model to generate reasoning trajectories and employ Qwen3-Next-80B-A3B model to extract visual annotations because of their powerful multimodal reasoning and language understanding capabilities. The prompts used in data collection and collection pipeline are same as those in the paper. We name the model trained on this dataset "Perception-R1-Mulberry-7B". The experimental results are as follows:
>
> |Model Name|MathVista|MathVerse|MathVision|WeMath|MMMU|MMMU-Pro|MMStar|EMMA|Avg|
> |:-|:-:|:-:|:-:|:-:|:-:|:-:|:-:|:-:|:-:|
> |Qwen2.5-VL-7B-Instruct|68.1|47.4|25.1|61.4|55.2|36.4|60.2|24.9|47.3|
> |Qwen2.5-VL-7B-Instruct + GRPO on Geometry3k|73.3|51.3|26.6|69.5|58.0|38.2|63.1|24.9|50.6|
> |**Perception-R1-7B**|74.2|54.3|28.6|72.0|60.8|42.4|64.5|27.5|53.0|
> |Qwen2.5-VL-7B-Instruct + GRPO on Mulberry|72.6|46.2|27.8|66.8|52.1|42|62.1|26.4|49.5|
> |**Perception-R1-Mulberry-7B**|73.4|51.2|27.1|69.9|59.1|42.2|62.6|27.2|51.6|
>
> From the table, we can see that **Perception-R1-Mulberry-7B still outperforms standard GRPO by 2.1 points on average across all benchmarks, demonstrating the effectiveness of our method.** We believe the reason why Perception-R1-7B outperforms Perception-R1-Mulberry-7B is that the collected Mulberry data lacks math reasoning content (especially geometry) and mainly focuses on pure visual perception, which leads to worse performance on math benchmarks.
>
> We will include this experiment in Appendix B.3 to provide a broader generalization analysis of our method in the revised version of our paper.
>
> [1] Mulberry: Empowering mllm with o1-like reasoning and reflection via collective monte carlo tree search

---

> > ### Comment · Reviewer_5AFL · 2025-11-25
> >
> > The authors' response is greatly appreciated. I will keep my rating as 8.

---

### Official Review · Reviewer_2S2F · 2025-10-31

**Soundness:** 4
**Presentation:** 3
**Contribution:** 3
**Rating:** 6
**Confidence:** 3

**Summary:**

This paper introduces Perception-R1, a reinforcement learning framework that enhances multimodal reasoning in MLLMs by explicitly improving their visual perception. Specifically, the authors (1) extract visual annotations from correct chain-of-thought trajectories as ground-truth perceptual references, (2) employ a judging LLM to evaluate the consistency between these annotations and the model’s generated reasoning, and (3) aggregate this feedback with accuracy and format rewards under the GRPO optimization scheme.

**Strengths:**

- The idea of augmenting RL with a verifiable visual perception signal represents a clear conceptual advance over prior RLVR frameworks (e.g., Vision-R1, MM-Eureka) that focus solely on final answer correctness.
- The authors conduct extensive evaluations on multiple multimodal benchmarks, demonstrating the method's effectiveness and robustness.
- The paper is well-structured and clearly written.

**Weaknesses:**

- The paper lacks systematic exploration of critical parameters such as the perception reward weight (γ) and judgment thresholds, leaving robustness questions unanswered.
- Although data-efficient, the additional judging and reward assignment stages may increase computational overhead, which is not quantitatively discussed.
- The paper would benefit from more qualitative evidence demonstrating how the model’s perception improves—e.g., visual attention maps, step-by-step perception-reasoning examples, or case studies showing corrected misperceptions. Such analyses would strengthen interpretability and directly connect the proposed reward to perceptual behavior.

**Questions:**

- The method’s success relies heavily on the quality and alignment of the judging LLM used to evaluate perceptual consistency. As shown in Figure 3(b–c), weaker judges introduce reward hacking and degrade performance, but the paper stops short of analyzing why this happens or proposing safeguards (e.g., calibration, ensemble judgment, or confidence filtering). Further discussion or mitigation strategies would make the approach more robust and reproducible across settings.
- The perception reward weight (γ) and the number/quality of visual annotations are central to the method, yet their interactions are not fully studied. Figure 3(a) provides only coarse exploration. More systematic experiments varying γ and annotation noise would clarify stability and guide practitioners in tuning the method.

---

> ### Author Response · Authors · 2025-11-22
> **[Part 1/2] Author Response**
>
> We sincerely thank you for your insightful suggestions on our experiments. Here are our responses to your concerns:
>
> ---
>
> **W1**: Lack of systematic exploration of perception reward weight ($\gamma$) and judgment thresholds.
>
> **A**: Thank you for your suggestion. Regarding the perception reward weight ($\gamma$), in addition to the average performance shown in Figure 3(a), we present below the performance of models trained with different $\gamma$ values on all benchmarks:
>
> |$\gamma$|MathVista|MathVerse|MathVision|WeMath|MMMU|MMMU-Pro|MMStar|EMMA|Avg|
> |:-|:-:|:-:|:-:|:-:|:-:|:-:|:-:|:-:|:-:|
> |0.0 (GRPO)|73.3|51.3|26.6|69.5|58.0|38.2|63.1|24.9|50.6|
> |0.1|72.7|54.1|28.5|70.9|60.0|41.2|65.4|27.8|52.6|
> |0.3|73.0|54.4|29.0|71.7|60.5|42.6|63.7|28.1|52.9|
> |0.5|75.5|53.0|27.6|70.5|59.1|42.9|65.5|27.4|52.7|
> |**0.7 (Perception-R1-7B)**|74.2|54.3|28.6|72.0|60.8|42.4|64.5|27.5|53.0|
> |0.9|72.4|53.7|28.4|72.2|60.9|40.7|64.1|28.0|52.5|
>
> From the table, we can notice that **the average performance of models trained with different $\gamma$ values (except 0.0) are pretty close, while all of them significantly surpass standard GRPO, demonstrating the robustness and effectiveness of our training framework.**
>
> As for the judgment thresholds, we believe there may be some misunderstandings. In our framework, there is no need for an extra hyperparameter to determine the consistency between the atomic visual annotations and the policy model’s rollouts. The judgment result for each atomic visual annotation can only be a binary value (0 or 1), which is directly produced by the judging LLM. The prompt for judging can be found in Figure 8, page 26 in the paper.
>
> ---
>
> **W2**: Computational overhead analysis about the proposed training pipeline.
>
> **A**: Actually, we have conducted a detailed computational overhead analysis in Appendix B.6 in the paper. We categorize the computational overhead into data preparation and training time, and estimate the computational overhead of all baselines and Perception-R1 as follows:
>
> |Model|Data Preparation Cost (#Tokens)|Training Time Cost (GPU-Hours)|
> |:-|:-|:-|
> |**Perception-R1**|1.1M Tokens|167.4 A800-Hours (1.4K RL)|
> |Vision-R1|134M Tokens|3392 H800-Hours (200K SFT+10K RL)|
> |MM-Eureka|0|>167.4 A800-Hours w.h.p (15K RL)|
> |SophiaVL-R1|34.9M Tokens|>167.4 A800-Hours w.h.p. (158K SFT+130K RL)|
> |VLAA-Thinker|29.6M Tokens|<167.4 A800-Hoursw.h.p. (25K RL)|
> |OpenVLThinker|About 5.7M Tokens|>167.4 A800-Hoursw.h.p. (25K SFT + RL)|
> |R1-Onevision|>1.1M Tokens w.h.p|>167.4 A800-Hoursw.h.p. (155K SFT + 10K RL)|
> |R1-VL|0|>167.4 A800-Hours w.h.p. (260K SFT + 10K RL)|
>
> From the table, we can see that **the overall computational overhead of Perception-R1 is still among the smallest.** The additional computational overhead arises from data preparation and GPU hours for judgment, consuming 1.1M more tokens and 30.2% more GPU hours compared to standard GRPO. Due to character limitations, please kindly refer to Appendix B.6 in the paper for the detailed estimation process.
>
> ---
>
> **W3**: Qualitative evidence demonstrating how the model’s perception improves.
>
> **A**: We have provide some case evidences from MathVista benchmarks in Appendix B.7. To provide better qualitative evidences, we will further include more cases from multimodal general benchmarks such as MMMU-Pro and MMStar in the revised version of our paper.

---

> ### Author Response · Authors · 2025-11-22
> **[Part 2/2] Author Response**
>
> **Q1**: Further analysis about why weaker judges lead to degrade performance, and safeguard to prevent it.
>
> **A**: Thank you for your insightful question!
>
> - **The analysis of this phenomenon**:
> Under the ideal circumstances, the judging LLM will assign 1 to the correct image description and assign 0 to erroneous ones in policy model's rollouts. Therefore, the rollouts containing more correct image descriptions will be ranked higher and encouraged during RL, while those without or even fewer correct descriptions will be suppressed because of the group mean mechanism of GRPO. However, in practical situations, there exists 2 kinds of errors made by judging LLM: **1. failing to recognize correct description in the rollout (1 -> 0)**, and **2. misjudging an incorrect description as correct (0 -> 1)**. The first type of errors does not lead to model collapse. In the worst case, it reduces to standard GRPO, where the visual perception reward does not exist. The reason why the model judged by Qwen2.5-7B-Instruct performs even worse than the base model on MathVerse and MathVision in Figure 3(b) is attributed to the second type of error. Since the base model itself may produce erroneous image descriptions during RL, this second kind of judging error encourages such erroneous behavior and suppresses any potentially correct descriptions, ultimately severely harming the model’s capabilities. We can see in Figure 3(b) that the visual perception reward produced by Qwen2.5-7B-Instruct climbs to a high value (>0.8) much more quickly than that of the Qwen2.5-32B-Instruct model. This indicates that many second-type judging errors are occurring, which ultimately leads to poor performance.
> - **Safeguards to preventing this phenomenon**:
> This is a valuable direction to further improve the overall training pipeline. In fact, we have tried to prevent the occurance of above 2 kinds of judging errors by scaling up the judging LLM, which will bring more accurate and robust judging results. The ensemble judgment you mentioned essentially enhances the correctness and robustness of the judging results, which is similar to our approach. As for calibration and confidence filtering, which aims to discard potentially erroneous judgments at inference time, we consider them promising directions, but leave them for future work due to their additional computational overhead.
>
> ---
>
> **Q2**: More systematic experiments varying $\gamma$ and annotation noise.
>
> **A**: Thank you for your suggestion. The detailed performance of models trained with varying $\gamma$ values on all benchmarks are presented in W1. Here we present experimental results with noisy annotations. In our experimental settings, we simulate the negative effects of noisy annotations by randomly flipping the judgments produced by the judging LLM at a fixed proportion.
>
> The performance of Perception-R1 with 10% and 20% random flipped visual perception reward are as follows:
>
> |Random flip ratio|MathVista|MathVerse|MathVision|WeMath|MMMU|MMMU-Pro|MMStar|EMMA|Avg|
> |:-|:-:|:-:|:-:|:-:|:-:|:-:|:-:|:-:|:-:|
> |Qwen2.5-VL-7B-Instruct + GRPO|73.3|51.3|26.6|69.5|58.0|38.2|63.1|24.9|50.6|
> |**0% (Perception-R1-7B)**|**74.2**|**54.3**|28.6|**72.0**|**60.8**|42.4|**64.5**|27.5|**53.0**|
> |10%|72.2|51.1|**29.1**|69.2|60.5|**42.9**|63.1|**28.3**|51.9|
> |20%|70.0|50.6|27.4|70.7|60.9|42.0|62.1|27.9|51.5|
>
> We can see that **even with 20% of visual perception reward corrupted, the average performance of the model still surpasses standard GRPO, demonstrating the robustness of our method.** Notably, the performance degradation mainly comes from MathVista and MathVerse. This may be because these two benchmarks contain a large number of geometry test cases that are similar to our training data. **In general benchmarks like MMMU and MMMU-Pro, the model trained with corrupted annotations still performs on par with Perception-R1-7B, further demonstrating the robustness of our training pipeline.**
>
> We will include the experiments in W1, Q2 and the analysis in Q1 in Appendix B to discuss the robustness of our method in the revised version of our paper.

---

> ### Comment · Reviewer_2S2F · 2025-11-27
>
> I appreciate the authors’ detailed response, which has addressed my previous concerns.

---

### Official Review · Reviewer_cpnq · 2025-11-01

**Soundness:** 2
**Presentation:** 3
**Contribution:** 3
**Rating:** 4
**Confidence:** 4

**Summary:**

This paper tackles a key but often neglected limitation in reinforcement learning for Multimodal Large Language Models (MLLMs): existing Reinforcement Learning with Verifiable Rewards (RLVR) methods focus solely on final answer correctness, overlooking the accuracy of visual perception during reasoning. The authors show that such outcome-only rewards allow models to guess correct answers despite severe perception errors. To address this, they propose Perception-R1, which introduces a verifiable visual perception reward into RLVR. This reward is derived from textual visual annotations extracted from high-quality CoT trajectories and evaluated by a judging LLM that measures consistency between model outputs and these annotations.
Contributions:
1. Empirically and statistically demonstrate that accuracy-only RLVR fails to enhance multimodal perception.
2. Introduce a novel, verifiable visual perception reward that alleviates reward sparsity and improves perceptual grounding.
3. Achieve state-of-the-art performance on multiple multimodal reasoning benchmarks using only 1,442 training samples, showing exceptional data efficiency.

**Strengths:**

1. This paper reveals the impact of poor perception on reasoning performance. Current RLVR methods fail to enhance multimodal perception, which fundamentally limits the reasoning performance of MLLMs.
2. The introduced Perception-R1 framework incorporates a novel visual perception reward that significantly strengthens the visual understanding and reasoning capabilities of MLLMs, particularly in mathematical reasoning tasks.
3. Extensive experiments across multiple benchmarks verify that Perception-R1 substantially improves both perception and reasoning performance, achieving superior results even with highly limited training data.

**Weaknesses:**

1. The paper claims that it enhances the multimodal reasoning capabilities of MLLMs through improved perception. However, the presented results do not provide direct evidence that the observed performance gains stem specifically from enhanced perception. I suggest including an analysis or ablation that directly links perception improvement to the reasoning gains.
2. While the paper reports significant improvements on multimodal math benchmarks, these results primarily reflect reasoning performance rather than perception itself. To convincingly demonstrate perception enhancement, it would be helpful to include evaluations on dedicated perception-level benchmarks (e.g., BLINK, MMBench, MME, or similar datasets).
3. The method employs Gemini-2.5 Pro to generate CoT trajectories and uses an LLM to extract visual annotations, followed by GRPO training on these annotations. This pipeline closely resembles a distillation process from Gemini-2.5 Pro, which may primarily transfer reasoning knowledge rather than genuinely improving perception. It would strengthen the paper to disentangle and clarify whether the observed gains truly originate from improved perception rather than implicit reasoning distillation.

**Questions:**

1.After distilling the CoT trajectories from Gemini 2.5 Pro, could you clarify why an LLM is used to transform these trajectories into atomic statements? In particular, how does this approach differ from directly inputting the trajectory data into the LLM to evaluate the atomic statements?

---

> ### Author Response · Authors · 2025-11-22
> **[Part 1/2] Author Response**
>
> Thank you for your comprehensive review! We will address your concerns one by one as follows:
>
> ---
>
> **W1**: Analysis or ablation that directly links perception improvement to the reasoning gains.
>
> **A:** We have conduct McNemar's test to Perception-R1 (Lines 403-407: `we also conduct McNemar’s test on Perception-R1. We investigate the same 50 problems as presented in Section 4.1 and find that the numbers of discordant cases for multimodal perception are 2 and 10, respectively. As a result, the exact binomial variation of McNemar’s test (McNemar, 1947) yields exact p value of 0.04, below the 0.05 significance threshold, indicating that the multimodal perception capabilities of Perception-R1 is substantially improved compared to the original MLLM.`), demonstrating the its explicitly enhanced perception capability. To further demonstrate the performance gains are mainly from the perception enhancement, we evaluate Perception-R1 and baseline models on text-only benchmarks including MATH-500 and GPQA, which can reflect the logical reasoning capabilities of them.
>
> The results are as follows:
>
> |Model Name|MATH-500|GPQA|
> |:--|:-:|:-:|
> |Qwen2.5-VL-7B-Instruct|54.0|6.7|
> |R1-OneVision-7B|57.2|5.4|
> |OpenVLThinker-7B|56.8|5.4|
> |VLAA-Thinker-7B|66.6|7.4|
> |SophiaVL-R1-7B|65.2|8.0|
> |MM-Eureka-7B|65.6|6.3|
> |Vision-R1-7B|66.2|10.7|
> |**Perception-R1-7B**|65.2|6.9|
>
> From the table, we can notice that the logical reasoning capabilities are on par with baseline models, **so the performance gains on multimodal math and general benchmarks can be attributed to improvements in perception.**
>
> ---
>
> **W2**: Performance on dedicated perception-level benchmarks.
>
> **A**: Thank you for your suggestion! We further evaluate Perception-R1 and baseline models on BLINK and MME benchmarks. The results are as follows:
>
> |Qwen2.5-VL models|MME|BLINK|
> |:-|:-:|:-:|
> |Qwen2.5-VL-7B-Instruct|64.8|55.3|
> |R1-OneVision-7B|80.1|48.7|
> |OpenVLThinker-7B|81.1|54.6|
> |VLAA-Thinker-7B|84.5|55.9|
> |SophiaVL-R1-7B|83.3|55.0|
> |MM-Eureka-7B|84.7|56.6|
> |Vision-R1-7B|84.5|51.3|
> |**Perception-R1-7B**|**85.4**|**58.0**|
>
>
> |Qwen2-VL models|MME|BLINK|
> |:-|:-:|:-:|
> |Qwen2-VL-7B-Instruct|64.2|51.7|
> |R1-VL-7B|65.9|51.2|
> |**Perception-R1-Qwen2-7B**|**84.0**|**53.4**|
>
> From the above tables, we can observe that **Perception-R1 and Perception-R1-Qwen2 also achieve the best performance on dedicated perception-level benchmarks MME and BLINK, demonstrating the effectiveness of perception enhancement.**
>
> ---
>
> **W3**: The improvement may come from the implicit reasoning of the distilled trajectory.
>
> **A**: Thank you for your constructive question. Actually, we had taken this issue into consideration in the early stage of our project. In our preliminary experiments, we manually checked the visual annotations extracted from the Geometry3K and GeoQA datasets respectively, and we rigorously classified each visual annotation into "visual information", "reasoning information", and "text information". Here, "visual information" refers to information that can only be obtained from the image; "reasoning information" refers to information that does not directly exist in the image or problem text, but can be obtained through reasoning; and "text information" refers to information that can be obtained directly from the problem text. We randomly sampled 30 data instances from the two datasets, and the statistical results of the classification are as follows:
>
> |Dataset|All annotations|Visual Information|Reasoning Information|Text Information|
> |:-|:-:|:-:|:-:|:-:|
> |Geometry3K|99|81 (82%)|10 (10%)|8 (8%)|
> |GeoQA+|114|29 (25%)|41 (36%)|44 (39%)|
>
> It is worth noting that any annotation that requires even mild reasoning to obtain is classified as "reasoning information" (e.g., Given an image of parallelogram ABCD, if the annotation states that "AB = CD", then it is classified as reasoning information). **Most "reasoning information" are such short statements rather than reasoning chains, minimizing the possibility of implicit reasoning distillation. Even with such a rigorous principle, the proportion of visual information in Geometry3K is still dominant (82%).** In comparison, the proportion is only 25% for the GeoQA+ dataset. This is the core reason why we chose the Geometry3K dataset as our training data, **as it provides better visual perception for our framework and isolates it from the influence of implicit reasoning.**
>
> In addition to the annotation classification results, the experiments on text-only benchmarks in W1 also showcase that the logical reasoning capabilities of Perception-R1 are on par with those of the baseline methods, **indicating that its performance superiority indeed stems from the improved visual perception.**
>
> We will include this preliminary experiment in Appendix B.2 in the revised version of our paper.

---

> > ### Comment · Reviewer_cpnq · 2025-11-27
> >
> > Thank you for the detailed additional experiments and clarifications.
> >
> > The newly provided evaluations on dedicated perception benchmarks (MME and BLINK), as well as the McNemar’s test and the vision-only subset analyses, convincingly demonstrate that the proposed method indeed brings substantial improvements in multimodal perception capability. These additions significantly strengthen the empirical support for the paper’s main claim regarding perception enhancement.
> >
> > However, despite these improvements, I still have reservations about the level of methodological novelty of the approach, particularly regarding the extent to which the gains are attributable to distillation from a stronger model (Gemini 2.5 Pro) rather than a fundamentally new learning mechanism. While the authors have attempted to mitigate implicit reasoning in the distilled annotations, the dependence on LLM-generated trajectories continues to leave some uncertainty about the originality of the contribution.
> >
> > In light of the remaining concerns about novelty and generalization, I will keep my score at a 4. I appreciate the authors’ efforts in expanding the analysis, and I hope the work continues to improve.
> >
> > Best wishes for the further development of this research.

---

> ### Author Response · Authors · 2025-11-22
> **[Part 2/2] Author Response**
>
> **Q1**: Why an LLM is used to transform these trajectories into atomic statements?
>
> **A**: The reasons why we chose to extract atomic visual annotations from trajectories rather than directly use them as reference are two folds:
>
> 1. **Removing implicit reasoning process.** As we discussed in W3, eliminating the influence of implicit reasoning in the distilled trajectories is also one of our core objectives, so that we can provide better visual references for computing the visual perception reward. Transforming entire trajectories into atomic statements can effectively remove the reasoning processes embedded in the trajectory, retaining only the statements and relationships between visual elements.
> 2. **Providing more accurate and fine-grained visual perception reward.** During RL, the rollouts of the base model may contain different numbers of erroneous image descriptions, so we need to accurately rank them, meaning the reward cannot be a simple binary value. To assign a fine-grained reward, directly applying an LLM to generate a float value between 0 and 1 based on entire trajectory would introduce the influence of implicit reasoning and lead to reward hacking (as discussed in Appendix B.5). Taking all factors into consideration, we ultimately chose to first extract the visual annotations and then compute the visual perception reward by checking the consistency between each atomic statement and the rollout one by one.

---

> ### Author Response · Authors · 2025-11-28
>
> We are very glad to hear that most of your concerns in the original review have been satisfactorily addressed. For your new concerns in the comment, we clarify them one by one:
>
> ---
>
> > the gains are attributable to distillation from a stronger model (Gemini 2.5 Pro) rather than a fundamentally new learning mechanism.
>
> We respectfully clarify that **the introduced Perception-R1 framework is indeed a new learning mechanism, and the performance gains are truely from the proposed framework.**
>
> Regarding the novelty of our framework, **the Perception-R1 training framework is an explicit improvement over GRPO for multimodal reasoning, aimed at enhancing the multimodal perception capabilities of MLLMs through a novel visual perception reward. This constitutes a new learning mechanism that is distinct from existing SFT and RL approaches.** The novelty of our method has also been acknowledged by you(Strengths 2:`...incorporates a novel visual perception reward...`), Reviewer 2S2F(Strengths 1:`augmenting RL with a verifiable visual perception signal represents a clear conceptual advance over prior RLVR frameworks`) and Reviewer 5AFL(Strengths 2:`The proposed visual perception reward is intuitive and cleverly designed...directly targets the identified bottleneck.`).
>
> **Regarding to the source of performance gains, we have conducted ablation studies in Table 2 to compare SFT and GRPO baselines using the same data, where the results prove that the performance gains are truly from our new learning mechanism (especially compared to SFT baseline).**
>
> ||MathVista|MathVerse|MathVision|WeMath|MMMU|MMMU-Pro|MMStar|EMMA|Avg|
> |:-|:-:|:-:|:-:|:-:|:-:|:-:|:-:|:-:|:-:|
> |Qwen2.5-VL-7B-IT + SFT|67.3|39.1|21.3|49.1|52.8|35.2|59.6|**28.3**|44.1|
> |Qwen2.5-VL-7B-IT + GRPO|73.3|51.3|26.6|69.5|58.0|38.2|63.1|24.9|50.6|
> |Perception-R1-7B|**74.2**|**54.3**|**28.6**|**72.0**|**60.8**|**42.4**|**64.5**|27.5|**53.0**|
>
> ---
>
> > the dependence on LLM-generated trajectories continues to leave some uncertainty about the originality of the contribution.
>
> We agree that the Perception-R1 training framework requires additional visual signals to implement the visual perception reward. However, what we fundamentally need are **accurate descriptions of the image** (we have mitigated the influence of implicit reasoning in the resposne to W1), and using an LLM to generate such descriptions is only one possible approach. As noted in Lines 285–286(`these CoT trajectories can also be obtained from existing open-source multimodal SFT datasets.`), **these trajectories can also be sourced directly from existing multimodal datasets.** Because all visual annotations in our method are extracted directly from solution trajectories rather than deliberate outputs, the proposed training pipeline can be naturally extended to any multimodal SFT dataset with accurate CoT trajectories, thereby ensuring the generalizability of our framework.
>
> **Moreover, in an era where synthesizing training data with powerful LLMs has become a common approach, we do not agree this undermines the originality of our method.** Baseline approaches including Vision-R1, R1-OneVision, and VLAA-Thinker require extensive SFT before RL, and their large-scale SFT data are also generated by strong LLMs (e.g., DeepSeek-R1 for Vision-R1 and VLAA-Thinker). In contrast, **Perception-R1 can be trained solely during the RL stage, yet still outperforms these baselines while requiring substantially lower data generation costs and training costs** (please kindly refer to the computational estimation in our response to Reviewer 2S2F’s W2 or Table 13 in the revised paper). **This further highlights the effectiveness and contribution of the Perception-R1 training framework.**
>
> We thank the reviewer's thorough concerns. If you have further concerns or still have any "uncertainty", please feel free to contact us at any time. We are looking forward to further discussions with you. :)

---

### Meta-Review · Area_Chair_KcCj · 2025-12-29

**Summary:**

The paper initially receives scores of 4, 6, 8, and 6. The AC considers that the reviewers’ main concerns are well addressed in the rebuttal (the first four concerns in the Reviewer Concerns Section). Although the method is not a fundamentally new learning mechanism, it has a reasonable motivation, and the experiments demonstrate the benefits of the perception rewards. Therefore, the AC recommends acceptance. The AC also suggests that the authors carefully revise the paper based on the rebuttal and add experiments using CoT trajectories generated from other models.

**Reviewer Concerns:**

The reviewers raised several concerns, with the most important ones summarized below.

1. Reviewers cpnq and 2S2F: concerns whether the reported performance gains truly come from improved visual perception. The rebuttal adds experiments on text-only reasoning benchmarks as well as perception-level benchmarks, providing direct evidence that the improvements are indeed linked to enhanced perception.
2. Reviewer 2S2F: concerns about hyperparameter sensitivity and the additional computational overhead.  The rebuttal included further experiments and analysis for these.
3. Reviewer 5AFL: asks for generalization analysis beyond geometry data . The rebuttal introduces perception-focused training data from Mulberry and showed improvements over the GRPO baseline.
4. Reviewer keG9: questions whether the improvements mainly come from GRPO rather than the proposed perception reward and raises concerns about robustness to CoT trajectory quality. The rebuttal provides GRPO-only baselines and perturbation experiments, demonstrating the additional benefit of the proposed rewards.
5. Reviewer cpnq: raises concerns about novelty and generalization, questioning whether the observed gains mainly result from distillation from a stronger model such as Gemini 2.5 Pro rather than from a fundamentally new learning mechanism.
6. Reviewer keG9: suggests using the same LLM to generate CoT trajectories as Vision-R1 to ensure a fair comparison.

The AC considers that the first four concerns are supported by additional experiments in the rebuttal and are well addressed. The fifth and sixth concerns remain, as the submission relies heavily on Gemini 2.5 Pro to generate CoT trajectories, and the impact of using trajectories from other models should be discussed. The AC recommends addressing this point in the camera-ready version.

Nevertheless, since the proposed perception rewards consistently improve over GRPO when using the same trajectories, the AC considers the core contribution of Perception-R1 to be validated.

**Reviewer Scores:**

The reviewers keep their scores. Reviewers cpnq, 2S2F, and 5AFL explicitly state this position. Reviewer keG9 considers most concerns to be addressed, with one remaining question, but maintains an overall positive stance.

---

### Decision · Program_Chairs · 2026-01-26

Accept (Poster)